# RESIDUALVIT FOR EFFICIENT ZERO-SHOT NATURAL LANGUAGE TEMPORAL VIDEO GROUNDING

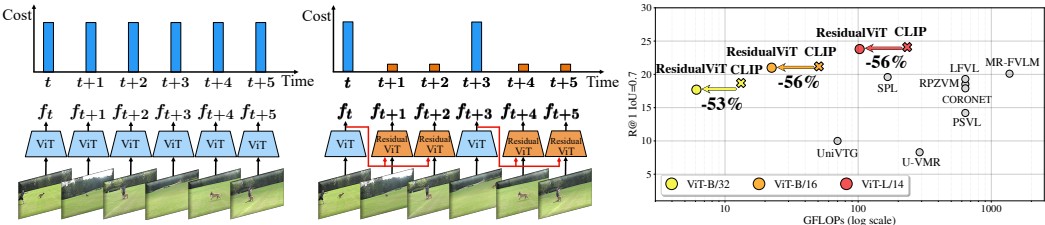

(a) Naive video encoding.  (b) Our ResidualViT video encoding.  (c) Comparison with state of the art.

Figure 1: **Efficient video feature encoding for Zero-shot Natural Language Temporal Video Grounding (NLTVG).** (a) The naive video encoding approach has a high computational cost when computing dense frame features, which are crucial for precise temporal grounding. (b) Our ResidualViT significantly reduces this cost, enabling both efficient and accurate temporally dense feature extraction. (c) Comparing with state-of-the-art methods, we achieve a striking balance between compute cost (x-axis) and accuracy (y-axis) for the NLTVG task (here on the Charades-STA dataset).

## ABSTRACT

The goal of this work is to efficiently compute frame-level features from videos for the Zero-Shot Natural Language Temporal Video Grounding (NLTVG) task. The contributions of this work are three-fold. First, we introduce a novel vision transformer (ViT) architecture, dubbed ResidualViT, that capitalizes on the large temporal redundancies in video. Our architecture incorporates (i) learnable residual connections that ensure temporal consistency across consecutive frames and (ii) a token reduction module for enhancing processing speed by selectively discarding temporally redundant information. Second, we describe a lightweight distillation strategy that enables learning parameters of ResidualViT from existing frame encoders without additional manual annotation. Finally, we validate the effectiveness of our approach across three diverse datasets, demonstrating significant reductions in computational cost (up to 60%) and improvements in inference speed (up to 2.5× faster), all while observing marginal accuracy reduction with respect to the teacher model.

## 1 INTRODUCTION

Video content has become ubiquitous across various platforms, driving the need for effective methods to parse and understand video data at scale. This is particularly crucial for applications such as search and retrieval, where the ability to quickly locate specific content within videos based on natural language queries can significantly enhance the user experience. Recent advancements in dual-encoder foundation models (Radford et al., 2021) have shown promising results in addressing these needs through zero-shot learning approaches, making them highly versatile. We argue that the zero-shot setting holds great merits as it favors developing a single strong model that can generalize to multiple tasks and datasets, enabling scalable, flexible, and easily maintained smart services for users and eliminating the need for task-specific finetuning. However, deploying such models in video understanding tasks, especially over extensive datasets, presents substantial computational challenges. Videos are notoriously data-heavy, and applying high-capacity models naively to every video frame is computationally prohibitive (Figure 1a). For instance, using CLIP's ViT-L/14 model to compute visual features for every frame in a 2.5M 30-second videos dataset on high-end A100

GPUs would require over 300 GPU days (or over 7000 GPU hours). Therefore, reducing the computational demands of current large-scale pretrained foundation models is imperative for enabling the practical and scalable deployment of video understanding technologies.

Prior approaches for reducing the compute cost of a pre-trained model primarily aim to distill a model's representation directly into a lower-capacity model (Dehghani et al., 2023; Hao et al., 2022; Heo et al., 2019; Wu et al., 2022b). While these efforts result in a more efficient model, distilling all the information from the larger model into the smaller one is challenging and often leads to a degradation in recognition accuracy. Moreover, these approaches naively treat video frames independently and do not explicitly take advantage of the temporal redundancy inherent in videos, which could further optimize processing.

To overcome these limitations, this work aims to compute video frame features efficiently given a pretrained vision transformer (ViT) model. As illustrated in Figure 1b, our solution capitalizes on the observation that nearby frames are often visually similar. Drawing inspiration from standard video compression techniques, which store a sparse set of *I-frames* (self-contained, fully-formed frames) and a denser set of *P-frames* (differences or changes from the previous frame), where the latter have high compression ratios (up to two orders of magnitude (Wu et al., 2018)), we adopt a similar strategy.

Our first contribution is an approach that computes the full ViT model representation on a sparse set of frames while providing an efficient approximation for representing the dense set of nearby frames. This strategy effectively mirrors the I-frame and P-frame method used in video encoding, leading to significant reductions in computational demand. We refer to the two sets of output representations as *I-features* (self-contained computed via a regular full ViT model) and *P-features* (efficiently computed using I-features and exploiting the temporal continuity of video). To compute the efficient P-features, we propose a novel vision transformer architecture (dubbed *ResidualViT*) that comprises two changes to the architecture of the pretrained ViT encoder. First, we compute a learnable *residual token* given a nearby I-feature. This residual token allows the ResidualViT encoder to exploit the temporal continuity of nearby video frames by incorporating their computed features. Second, we include a token reduction module (Ding et al., 2023; Haurum et al., 2023; Hou et al., 2022a; Bolya et al., 2022) in the ResidualViT encoder that significantly reduces the number of tokens used to compute P-features, substantially reducing their encoding costs. Combining these modules allows the ResidualViT encoder to efficiently and accurately approximate the target features.

As our second contribution, we propose a student-teacher distillation training objective that minimizes the loss between the vision-language embedding similarities produced by our efficient ResidualViT encoder and the features obtained from CLIP's pretrained Vision Transformer (ViT) backbone. This setup enables our ResidualViT encoder to replicate features from CLIP. The training is lightweight, as only the residual tokenizer module is learned while the ViT encoder weights remain frozen. This strategy allows us to fully harness the capabilities of CLIP without the need for large-scale training.

As our third contribution, we demonstrate the potential for practical and efficient search in videos provided by ResidualViT for the natural language temporal video grounding (NLTVG) task. Our model significantly reduces frame encoding costs with minimal search accuracy degradation (Figure 1c) on three diverse benchmarks. A thorough ablation study complements and validates our proposed solution. Lastly, ResidualViT's visual representations are tested on the complementary task of Automatic Audio Description generation (Han et al., 2023), achieving comparable performance to the CLIP baseline at a lower computational cost and demonstrating the strong generalization capabilities of our proposed architecture.

## 2 RELATED WORK

**Image Foundation Models for Video Applications.** The analysis of video data introduces many technical challenges arising from its inherent temporal and spatial complexities, large data volume, and high temporal redundancy. As a way to mitigate these challenges and ease the development of new tools, the research community has resorted to applying image-based models (He et al., 2016; Radford et al., 2021; Simonyan & Zisserman, 2014) to video tasks (Castro & Heilbron, 2022; Diwan et al., 2023; Luo et al., 2022; Nam et al., 2021; Soldan et al., 2022; 2021) with much success

despite the image-based architectures' inability to reason about the temporal dimension. Moreover, dedicated temporal modeling (Liu et al., 2023a; Ma et al., 2022; Tu et al., 2023; Xue et al., 2022) can offer potential accuracy gains at the expense of increased computational demands, highlighting a nuanced balance of efficiency and efficacy. Our work capitalizes on the CLIP image foundation model (Radford et al., 2021) to build an efficient video feature extraction framework that can be adopted for multiple downstream video tasks. We choose CLIP because of its excellent performance on multiple tasks (Radford et al., 2021; Shen et al., 2021; Lin et al., 2022) and native multi-modality (image and text), which can be adapted for video processing. Previous approaches leveraging CLIP for video tasks have utilized various strategies. These include applying temporal aggregation over frame representations (Buch et al., 2022; Luo et al., 2022; Ni et al., 2022), fine-tuning the model to capture motion patterns in videos (Castro & Heilbron, 2022; Weng et al., 2023), and employing carefully designed spatial and temporal adapters to harness the valuable pre-trained weights without modification (Lin et al., 2022; Pan et al., 2022; Yang et al., 2023; Park et al., 2023). Additionally, some methods have introduced prompt learning as a mechanism for domain adaptation (Ju et al., 2022). In a similar spirit, our work seeks to leverage pre-trained network weights without modification; however, we focus on reducing the computational cost of encoding individual frames by minimizing redundant temporal computations while preserving essential semantic details.

**Efficient Video Representations and Distillation.** Prior work has also looked at distilling into a lower-capacity model (Dehghani et al., 2023; Hao et al., 2022; Heo et al., 2019; Wu et al., 2022b) or developing efficient video representations for tasks such as semantic video segmentation (Liu et al., 2020b) or video recognition (Lin et al., 2019; Wu et al., 2022a; 2018). The former approaches result in a degradation of recognition accuracy due to the difficulty of distilling to a small model from a larger model. The latter approaches have investigated how to efficiently compute convolution in time (Lin et al., 2019), leverage the video compression representation in a convolutional network (Wu et al., 2018), or avoid computing the cross-attention in time for long videos (Wu et al., 2022a). Additionally, other methods tackle the efficient inference challenge through network pruning (Fang et al., 2023; Molchanov et al., 2016; He et al., 2017) reducing the number of parameters in convolutional networks and, consequently, the computational cost of pre-trained models. In contrast, we focus on the recent transformer-based ViT architectures (Dosovitskiy et al., 2020) (rather than convolutional models) that have demonstrated excellent scaling properties. Moreover, we focus on single-frame representations (such as CLIP (Radford et al., 2021)) that are often the video representation of choice for their versatility in large-scale practical setups involving natural language (Castro & Heilbron, 2022; Luo et al., 2022; Soldan et al., 2022), and consider the task of natural language video grounding, discussed next.

**Natural Language Temporal Video Grounding.** Natural language grounding in videos (Hendricks et al., 2017; Gao et al., 2017; Krishna et al., 2017) has emerged as a multi-modal generalization of the temporal activity localization task (Caba Heilbron et al., 2015) by replacing action classes with natural language sentences. Both tasks share common challenges, such as: (i) the annotation process is labor-intensive, which limits the size of benchmarks. (ii) The necessity for fine-grained temporal resolution demands dense frame sampling, resulting in significant computational requirements. To mitigate the annotation challenge, research has transitioned from conventional fully supervised methodologies (Barrios et al., 2023; Escorcia et al., 2019; Liu et al., 2020a; Mun et al., 2020; Soldan et al., 2021; Xu et al., 2023; Zeng et al., 2020; Zhang et al., 2020; Zhao et al., 2021) towards more flexible frameworks such as weak supervision (Chen et al., 2020; Huang et al., 2021; Zheng et al., 2022a;b) and zero-shot learning (Diwan et al., 2023; Gao & Xu, 2021; Holla & Lourentzou, 2023; Kim et al., 2023; Luo et al., 2024; Nam et al., 2021; Soldan et al., 2022; Wang et al., 2022a; Zheng et al., 2023). In a fully supervised setting, models are trained using videos, sentences, and temporal boundaries, while in weakly supervised approaches the temporal annotations are not to be available.

Closer to our research is the setup in which the textual or temporal labels are unavailable. In this setting, prior work has leveraged off-the-shelf concept detectors (objects, actions, and scenes) (Gao & Xu, 2021; Nam et al., 2021; Wang et al., 2022a) to automatically generate pseudo-annotations (sentence and temporal boundaries) on a target downstream task dataset and train a grounding model on such data. Other sources of pseudo supervision come from pretrained visual-language embedding spaces (Kim et al., 2023), commonsense sources (Holla & Lourentzou, 2023; Speer et al., 2017), and captioning methods (Zheng et al., 2023). Additionally, methods leverage complex proposal schemes based on feature clustering (Holla & Lourentzou, 2023; Kim et al., 2023; Nam et al., 2021) or sliding windows (Wang et al., 2022a), paired with strategies for supervised feature refinement. Although

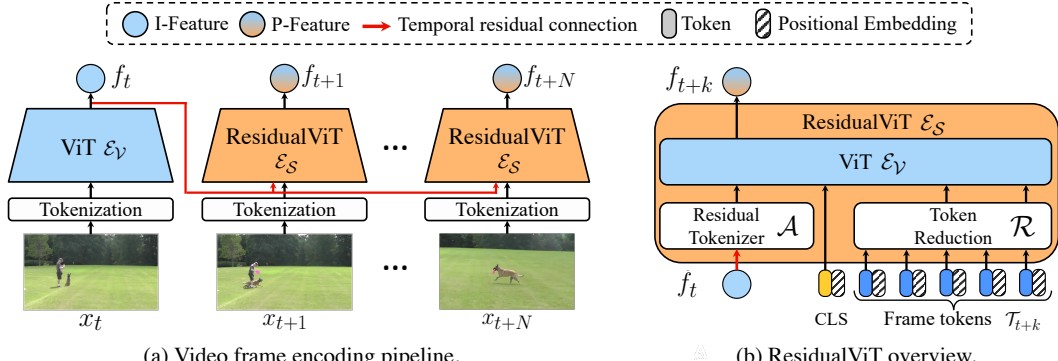

(a) Video frame encoding pipeline.          (b) ResidualViT overview.

Figure 2: **Model overview.** (a) Video frames are processed via two visual encoders $\mathcal{E}_\mathcal{V}$ and $\mathcal{E}_\mathcal{S}$ in an *interleaved* manner. For each frame encoded via the ViT $\mathcal{E}_\mathcal{V}$, $N$ subsequent frames are encoded using our lightweight ResidualViT $\mathcal{E}_\mathcal{S}$, significantly reducing the computational cost. (b) ResidualViT incorporates a token reduction module $\mathcal{R}$ for reducing the computation and the residual tokenizer $\mathcal{A}$ to ensure temporal consistency by propagating information from preceding frames.

these approaches do not use manually annotated labels, they still adapt (and learn) parameters on the training dataset for the target downstream task; therefore, we refer to them as *pseudo-supervised*. Differently from these works, we devise an all-purpose ResidualViT model for efficient video frame embedding computation without training on the downstream task dataset, which is effective for the task of natural language grounding in videos.

## 3 INTERLEAVED FEATURES FOR EFFICIENT NATURAL LANGUAGE TEMPORAL VIDEO GROUNDING

We investigate efficient video representation for the natural language temporal video grounding (NLTVG) task, which is formalized as follows: given an untrimmed video and a natural language query describing a specific moment, the goal is to predict the temporal span $(\tau_s, \tau_e)$ that corresponds to the described moment (Hendricks et al., 2017; Gao et al., 2017). This precise temporal moment localization requirement presents two major challenges. First, it requires the dense extraction of visual features from large volumes of video data. Second, it requires language and visual understanding to enable querying the model through natural language queries.

To address these challenges, we propose to adapt dual-encoder transformer-based pretrained models, focusing on making the visual encoder (ViT) (Dosovitskiy et al., 2020) more efficient for computing a temporally dense set of video features. Our approach capitalizes on the temporal redundancy inherent in videos, where consecutive frames often share redundant visual and semantic content as actions and scenes continuously evolve in time. This setting entails that naively encoding each frame independently leads to unnecessary computational overhead. In our study, we adapt the visual encoder from the dual-encoder vision-language model CLIP (Radford et al., 2021), which is well-known for its excellent performance in vision-language tasks. CLIP offers a versatile foundation for our visual encoder and provides a paired language encoder, allowing us to effectively model the nuanced visual-linguistic relationships needed for addressing the NLTVG task.

Figure 2a outlines our efficient visual encoding pipeline. Consider a video comprised of $n_v$ frames decoded at a constant frame rate, denoted as $\mathcal{X} = \{x_t\}_{t=1}^{n_v}$ with $x_t \in \mathbb{R}^{H \times W \times C}$, where $H$, $W$ and $C$ are the height, width, and number of channels of each frame. In alignment with standard vision transformer data processing, we convert each frame into a set of $K$ tokens, denoted by $\mathcal{T} = \{t_j\}_{j=1}^{K}$ with $t_j \in \mathbb{R}^d$, where $d$ is the token dimension. The embedding process for frame $x_t$ for $(t - 1) \bmod (N + 1) = 0$ consists of applying the visual encoder $\mathcal{E}_\mathcal{V} : \mathbb{R}^{|\mathcal{T}| \times d} \to \mathbb{R}^b$ on all frame tokens $\mathcal{T}_t$ to obtain an I-feature representation, $f_t = \mathcal{E}_\mathcal{V}(\mathcal{T}_t) \in \mathbb{R}^b$, where $b$ is the feature dimension. The subsequent $N$ frames $\{x_{t+k}\}_{k=1}^{N}$ are encoded using our ResidualViT encoder $\mathcal{E}_\mathcal{S} : \mathbb{R}^b \times \mathbb{R}^{|\mathcal{T}| \times d} \to \mathbb{R}^b$ to obtain P-features. Formally, we compute the P-features for those $N$ frames as $f_{t+k} = \mathcal{E}_\mathcal{S}(f_t, \mathcal{T}_{t+k}) \in \mathbb{R}^b$, where I-feature $f_t$ from frame $x_t$ is routed through the temporal residual connection (shown in red in Figure 2) to the ResidualViT encoder $\mathcal{E}_\mathcal{S}$ as temporal context. We define $N$ as the interleave factor, as it governs the interleaving of I-features and P-features. Note that in

our work, we use the output representation of the `[CLS]` token from the transformer architecture as our feature representation.

The following provides a detailed explanation of the design of our ResidualViT architecture (Sec. 3.1) and the associated training strategy (Sec. 3.2). Please see Appendix C for an in-depth discussion of our zero-shot watershed-based grounding algorithm, which operates atop both visual and language features.

## 3.1 RESIDUALVIT ARCHITECTURE

Our technical solution involves equipping the ViT encoder with two key components, as illustrated in Figure 2b: (i) *a token reduction module* ($\mathcal{R}$) and (ii) *a residual tokenizer module* ($\mathcal{A}$). The token reduction module selectively prunes input tokens to the ViT, retaining only the most informative ones, to **significantly reduce the encoding computational cost**. Concurrently, the residual tokenizer module propagates information from the last I-feature to the current P-feature **compensating for the information discarded** by the token reduction process.

Following standard ViT implementations, each frame $x_t$ is transformed into a set of patches $\{x_{(t,i)}\}_{i=1}^{|\mathcal{T}|}$ and projected to an embedding space $\mathbb{R}^d$, yielding a set of tokens $\mathcal{T}_t$. The number of patches and, therefore, the number of tokens $|\mathcal{T}|$ depends on the frame size ($H, W$) and the patch size ($P$): $|\mathcal{T}| = H \times W / P^2$. The tokens are given to the transformer encoder, which processes them with a self-attention mechanism, where the computational cost scales directly with the number of tokens. Reducing the token count can, therefore, save computation, but determining which tokens to safely discard to minimize loss of information remains a challenge.

In this work, we explore several token reduction strategies, including token dropping (or Patch-Dropout) (Ding et al., 2023; Haurum et al., 2023; Hou et al., 2022a; Liu et al., 2023b), which discards a subset of tokens based on a token dropping probability $p$. We also investigate token merging (Bolya et al., 2022), which progressively reduces the number of tokens at each transformer layer by a factor of $r$, and frame resolution reduction, which decreases the number of patches extracted from each frame. For detailed descriptions of the different token dropping strategies (*e.g.*, random, uniform, center, and motion-based), please refer to Appendix A. A comprehensive ablation study, presented in Appendix E, shows that token dropping offers the best trade-off between computational efficiency and model performance.

In our ResidualViT architecture, the token reduction module is used during both training and inference to reduce computational overhead. This setup implies that part of the visual information is discarded. Yet, thanks to the temporal redundancy of consecutive frames, we seek to exploit the semantic information present in the feature computed at time step $t$ to recover the missing spatial information induced by the token reduction operation at time step $t + k$. In detail, the ResidualViT architecture takes as input I-feature $f_t$ from frame $x_t$ via the temporal residual connection and transforms this feature into a residual token as $\mathcal{A}(f_t) \in \mathbb{R}^d$ via a learnable mapping $\mathcal{A} : \mathbb{R}^b \to \mathbb{R}^d$. This transformation is necessary to learn a token representation that is compatible with the visual encoder $\mathcal{E}_\mathcal{V}$ and can propagate useful information from the previous I-feature. The residual token is then concatenated with the `[CLS]` token and a small subset of frame tokens output by the token reduction module $\mathcal{R}(\mathcal{T}_{t+k})$. The resulting concatenated tokens are then fed into the visual encoder $\mathcal{E}_\mathcal{V}$ to obtain P-feature $f_{t+k}$. In our work, we implement the residual tokenizer $\mathcal{A}$ as a linear transformation. The addition of the residual token to the input of the transformer encoder adds a negligible computational overhead of about 0.1 GFLOPS (*i.e.*, $0.1\%$ of the frame encoding cost using the CLIP ViT-L/14 backbone). Despite the mapping $\mathcal{A}$ being a small linear layer, our solution is capable of providing informative cues even when most frame tokens are unavailable.

Following our design, when token dropping is used, the average embedding cost of our pipeline can be approximated as:

$$C = \frac{C_{\mathcal{E}_\mathcal{V}} + NC_{\mathcal{E}_\mathcal{S}}}{N + 1} \approx C_{\mathcal{E}_\mathcal{V}} \frac{1 + (1 - p)N}{N + 1}, \tag{1}$$

where $C_{\mathcal{E}_\mathcal{V}}$ and $C_{\mathcal{E}_\mathcal{S}}$ are the costs of encoding a frame using the visual encoder $\mathcal{E}_\mathcal{V}$ and $\mathcal{E}_\mathcal{S}$, respectively. Here, the interleave factor $N$ corresponds to the number of frames encoded by the ResidualViT with the reduced cost, and $p$ is the token reduction probability. It should be noted that when $N > 0$ and $p > 0$, the average embedding cost $C$ is strictly lower than $C_{\mathcal{E}_\mathcal{V}}$. For em-

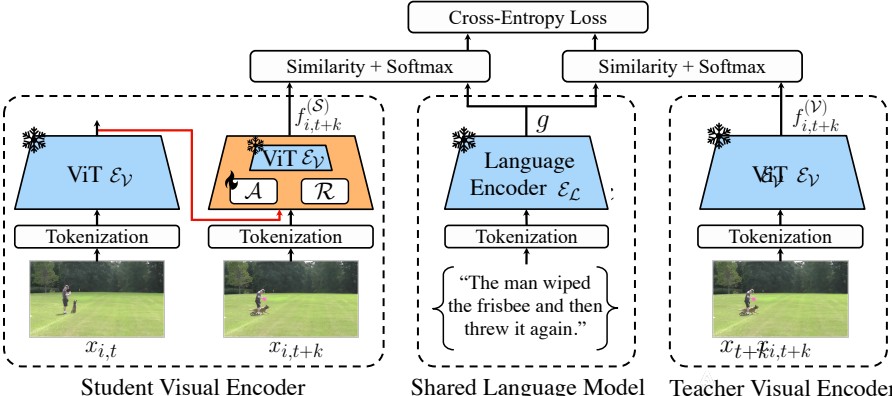

Figure 3: **ResidualViT training ($\mathcal{J}_{L \to V}$ loss).** We supervise the training of the residual token projection $\mathcal{A}$ via feature distillation. The loss encourages the output features of ResidualViT ($f_{i,t+k}^{\mathcal{S}}$) to approximate those of the pre-trained ViT encoder ($f_{i,t+k}^{\mathcal{V}}$).

pirical evidence demonstrating the reduction in wall-clock time for frame encoding when utilizing ResidualViT compared to a standard ViT, refer to Appendix F.

## 3.2 Training ResidualViT

The objective of our training is to supervise the residual tokenizer module $\mathcal{A}$, our only trainable component, such that the output frame feature computed by our ResidualViT $\mathcal{E}_{\mathcal{S}}$ closely approximates the feature computed via the original ViT encoder $\mathcal{E}_{\mathcal{V}}$ for the same frame. The challenge lies in the fact that the ViT encoder has access to every token $\mathcal{T}_{t+k}$ from the input frame while the transformer encoder of our ResidualViT only receives a sparse set of frame tokens due to the token reduction module $\mathcal{R}(\mathcal{T}_{t+k})$ together with the residual token $\mathcal{A}(f_t)$ (Figure 2b). We achieve this objective via feature distillation (Heo et al., 2019; Hinton et al., 2015; Ilharco et al., 2021), where the original foundation model serves as a "teacher" network while our ResidualViT acts as the "student" network. In our study, we leverage the powerful CLIP (Radford et al., 2021) foundation model to initialize the transformer encoders (*e.g.*, ViT-B/32, ViT-B/16, or ViT-L/14). We fully exploit the CLIP model by including its language encoder $\mathcal{E}_{\mathcal{L}}$ in the feature distillation pipeline and perform the training using paired video and language samples.

We illustrate the training process in Figure 3. Let $\mathcal{B} = \{(\mathcal{X}_i, \ell_i)\}_{i=1}^{B}$ be a batch of videos $\mathcal{X}_i$, and their corresponding textual descriptions $\ell_i$. From each video $\mathcal{X}_i$, we decode $N_{\text{Train}} + 1$ frames at a constant frame rate starting at time step $t$. These frames are then encoded via the ViT $\mathcal{E}_{\mathcal{V}}$ (teacher) and ResidualViT $\mathcal{E}_{\mathcal{S}}$ (student) and the corresponding features $f_{i,t+k}^{(\mathcal{V})}$ and $f_{i,t+k}^{(\mathcal{S})}$ are output for each time step $t + k$ for $k \in \{1, \ldots, N_{\text{Train}}\}$. Furthermore, let $g \in \mathbb{R}^{b \times B}$ be a matrix of features with dimension $b$ computed from all the textual descriptions $\{\ell_i\}$ in the batch using the language encoder $\mathcal{E}_{\mathcal{L}}$. We aim to train the ResidualViT encoder to match soft targets, which are the similarities between the ViT encoder (teacher) features and the language features. To achieve this goal, we optimize a cross-entropy loss over the softmax inner product between the vision features $f_{i,t+k}$ and language features $g$,

$$\mathcal{J}_{L \to V} = - \sum_{i=1}^{B} \sum_{k=1}^{N_{\text{Train}}} \sum_{j=1}^{B} \sigma_j \left( g^{\mathsf{T}} f_{i,t+k}^{(\mathcal{V})} \right) \log \left( \sigma_j \left( g^{\mathsf{T}} f_{i,t+k}^{(\mathcal{S})} \right) \right), \tag{2}$$

where $\sigma_j(x) = \exp(x_j) / \sum_c \exp(x_c)$ is the $j$-th component of the softmax function of vector $x$. Here, the sum over $c$ in the denominator of the softmax ensures that for a given image feature $f_{i,t+k}$ similarities to all language descriptions $g$ in the batch sum to one, converting them to a probability distribution. The inner sum in equation 2 sums over the language descriptions $j$ in the batch; the middle sum adds losses for all the frames $k$ in each video; and finally, the outer sum sums over all videos $i$ in the batch. Please note that due to the softmax normalization over the language features, the computation is asymmetric. Hence, we also define in an analogous manner a video to language loss $\mathcal{J}_{V \to L}$ where the sigmoid normalization is over the vision features in the batch.

The final loss is then the sum of the two losses. The overall learning problem is then formulated as the following minimization $\min_{\mathcal{A}} \left( \mathcal{J}_{L \to V} + \mathcal{J}_{V \to L} \right)$, where $\mathcal{A}$ are the parameters of the residual

| | Token Reduction | Interleave Factor | Residual Tokenizer (Distilled) | Charades-STA R@1 IoU=0.5 | IoU=0.7 | Avg. Cost Feature/sec (GFLOPs) |
|---|---|---|---|---|---|---|
| **a.** | | | | 42.9 | 24.1 | 233.4 |
| **b.** | ✓ | | | 28.5 | 14.5 | $35.7_{(-85\%)}$ |
| **c.** | ✓ | ✓ | | 38.9 | 22.8 | $102.0_{(-56\%)}$ |
| **d.** | ✓ | ✓ | ✓ | 41.5 | 23.8 | $102.6_{(-56\%)}$ |

Table 1: **Architecture ablation.** We ablate the main components of our architecture: the token reduction module, the interleave factor, and the distilled residual tokenizer. We set the token reduction probability $p$ to 85%, $N = 2$, and use the ViT-L/14 backbone.

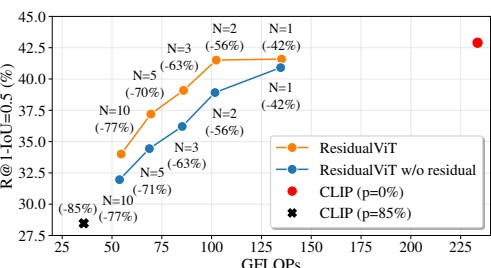

Figure 4: **Interleaving frames** ($N$). The cost vs. performance trade-off for varying N. Our ResidualViT (**orange**) almost retains CLIP's (**red**) performance for $N = 1$ and 2 while reducing cost by 56%.

tokenizer module. Please note that this loss not only encourages the visual representation of the two models to be close to each other but also supervises the feature distillation to preserve the joint vision-language space of the original CLIP model as the language features are shared between the teacher ViT encoder and the student ResidualViT encoder. We optimize the loss over samples from a paired video-language dataset. Please note that as we are learning (distilling) only a small number of parameters of the residual tokenizer $\mathcal{A}$, which is a single linear layer, our learning formulation does not require huge training datasets often required in typical distillation set-ups when an entire large model is distilled into another (smaller) model.

## 4 EXPERIMENTS

**Evaluation Metrics.** Following (Hendricks et al., 2017; Gao et al., 2017), grounding accuracy is measured via Recall@$K$ for IoU=$\theta$ with $K$=1 and $\theta \in \{0.5, 0.7\}$. The computational cost of video encoding is measured in GFLOPs, reflecting the average cost per second based on the frame rate and the cost to encode a single frame. We direct the reader to Appendix G for complementary information on the metrics, Appendix H for implementation and distillation training details, and Appendix C for the presentation of the zero-shot grounding algorithm and inference details.

**Evaluation Datasets.** We evaluate our approach on the Charades-STA (Gao et al., 2017), ActivityNet-Captions (Krishna et al., 2017) and MAD (Soldan et al., 2022) datasets.

The Charades-STA dataset is built atop the Charades dataset (Sigurdsson et al., 2016) and consists of unedited videos of human activities that follow predefined scripts. We evaluate on the testing set (1334 videos and 3720 textual annotations). The ActivityNet-Captions dataset is built atop the ActivityNet dataset (Caba Heilbron et al., 2015) and comprises edited videos scraped from the internet containing a clear taxonomy of human activities, augmented with temporally grounded language descriptions. We evaluate on the val-02 split (4885 videos and 17031 sentences). The MAD dataset, based on audio descriptions from movie data, consists of long videos with an average duration of 110 minutes. We report performance on the test set, which includes over 72K sentences grounded in 112 movies. As we operate in a zero-shot manner, we do not utilize the training sets of the above datasets in this study.

### 4.1 ABLATION STUDY

In this section, we perform multiple ablations to assess the impact of our design choices. We report performance on the Charades-STA dataset using the ViT-L/14 backbone. When token reduction is used, we employ the motion-based strategy (Appendix A) with probability $p = 85\%$. For all experiments that interleave frames, we set $N$=2. Five additional ablations are detailed in Appendix E. These ablations investigate (i) the token drop strategy for token reduction, (ii) token drop probability, (iii) the adoption of token merging for token reduction, (iv) the impact of frame resolution reduction as an alternative to token reduction and (v) replacing the distillation objective.

**Architecture Ablation.** In Table 1, we analyze the contribution of the main architecture components of our model to downstream task performance. With an average frame encoding cost of 233.4 GFLOPs, the CLIP baseline (**a.** in Table 1) establishes our upper bound grounding accuracy. When we apply token reduction across all frames (**b.**), we observe an 85% decrease in computational cost.

| | Supervision | Use Downstream Task Data | Charades-STA IoU=0.5 | IoU=0.7 | Avg. Cost Feature/sec (GFLOPs) | ActivityNet Captions IoU=0.5 | IoU=0.7 | Avg. Cost Feature/sec (GFLOPs) |
|---|---|---|---|---|---|---|---|---|
| 2D-TAN (Zhang et al., 2020) | Full | ✓ | 39.8 | 23.3 | **74.2** | 44.0 | 27.4 | **19.3** |
| CPNet (Li et al., 2021) | Full | ✓ | 60.3 | 38.7 | 638.3 | 40.6 | 21.6 | 38.5 |
| CRaNet (Sun et al., 2023) | Full | ✓ | **60.9** | **41.3** | 296.8 | **47.3** | **30.3** | **19.3** |
| WSTG (Chen et al., 2020) | Weak | ✓ | 27.3 | 12.9 | **38.5** | 23.6 | – | 38.5 |
| CRM (Huang et al., 2021) | Weak | ✓ | 34.8 | 16.4 | 638.3 | 32.2 | – | **23.2** |
| CPL (Zheng et al., 2022b) | Weak | ✓ | **49.2** | **22.4** | 445.2 | **31.4** | – | 115.5 |
| U-VMR (Gao & Xu, 2021) | Pseudo | ✓ | 20.1 | 8.3 | 289.5 | 26.4 | 11.6 | 962.5 |
| PSVL (Nam et al., 2021) | Pseudo | ✓ | 31.3 | 14.2 | 638.1 | 30.1 | 14.7 | 38.5 |
| PZVMR (Wang et al., 2022a) | Pseudo | ✓ | 33.2 | 18.5 | 638.1 | 31.3 | **17.8** | 38.5 |
| CORONET (Holla & Lourentzou, 2023) | Pseudo | ✓ | 34.6 | 17.9 | 638.1 | 28.2 | 12.8 | 38.5 |
| LFVL (Kim et al., 2023) | Pseudo | ✓ | 37.2 | 19.3 | 638.1 | **32.6** | 15.4 | 38.5 |
| SPL (Zheng et al., 2023) | Pseudo | ✓ | **40.7** | **19.6** | **166.5** | 27.2 | 15.0 | 83.3 |
| UniVTG (Lin et al., 2023) | Zero-Shot | ✗ | 25.2 | 10.0 | 70.0 | – | – | – |
| MR-FVLM (Luo et al., 2024) | Zero-Shot | ✗ | **42.9** | 20.1 | 1370.0 | 27.9 | 11.6 | 370.0 |
| CLIP (B/32) | Zero-Shot | ✗ | 35.9 | 18.7 | 13.2 | 27.8 | **13.9** | 4.4 |
| ResidualViT (B/32) (ours) | Zero-Shot | ✗ | 34.2 | 17.7 | $6.1_{(-53\%)}$ | 27.3 | 13.7 | $2.0_{(-53\%)}$ |
| CLIP (B/16) | Zero-Shot | ✗ | 37.7 | 21.2 | 50.7 | 28.1 | 13.8 | 16.9 |
| ResidualViT (B/16) (ours) | Zero-Shot | ✗ | 37.8 | 21.0 | $22.4_{(-56\%)}$ | 27.5 | 13.8 | $7.5_{(-56\%)}$ |
| CLIP (L/14) | Zero-Shot | ✗ | **42.9** | **24.1** | 233.4 | **29.1** | 13.8 | 77.8 |
| ResidualViT (L/14) (ours) | Zero-Shot | ✗ | 41.5 | 23.8 | $102.6_{(-56\%)}$ | 28.3 | 13.5 | $34.2_{(-56\%)}$ |

Table 2: **Short video state-of-the-art comparison.** We compare our approach against state-of-the-art methods using different levels of supervision. Our ResidualViT reduces the cost of frame encoding by 56% while closely retaining the performance of the CLIP model. The best method in each block of directly comparable methods is bolded, and the second-best method is underlined.

However, this setting induces marked absolute declines in grounding accuracy of 14.4% and 9.6% in our metrics, which translates to a relative drop of 34−40%. The introduction of our interleave strategy (**c.**), which alternates encoding one frame without token reduction and $N$ frames with token reduction (where $N = 2$), shows an increase in grounding accuracy of 10.4% and 8.3% while only using 44% of the original computational budget, which is a first step in closing the grounding accuracy gap with respect to the target performance. Compared to the full CLIP model, the grounding accuracy drop narrows to a modest 4 and 1.3 percentage points (5−9% relative drop), yet this configuration only incurs 44% of the original computational cost. Further, adding the residual tokenizer learned via distillation (**d.**) comes at a negligible compute cost but further boosts grounding accuracy closer to the target CLIP model, showing only a minor ∼1% absolute drop.

**Interleave Factor $N$ and Benefits of Distillation.** In Figure 4, we explore the relationship between grounding accuracy and computational cost as we vary the number of interleaved frames ($N$). In this visualization, the baseline CLIP model is shown in **red**, while our ResidualViT, applied with and without the distilled residual tokenizer module, is shown in **orange** and **blue**, respectively. We vary $N \in \{1, 2, 3, 5, 10\}$. When setting $N = 1$, grounding accuracy is marginally impacted, yet a large computational cost reduction is already achieved (42%). Notably, we see a further accuracy drop when the residual tokenizer is removed (**blue**), demonstrating the importance of the distillation training. At $N = 2$, the cost savings increases to 56% with virtually no accuracy change for ResidualViT. However, the importance of the learnable residual connection (via the residual token learnt by the distillation training) becomes more evident as the difference between the two configurations widens with substantial drops when the residual token is not employed. Increasing $N$ beyond this point sees diminishing returns in cost savings, now at 63%, and a noticeable decrease in accuracy. This decline is attributed to the growing temporal gap between I-features and P-features, leading to a weakened visual correlation and, thus, reduced efficacy. We regard $N = 2$ as the best trade-off.

## 4.2 COMPARISON WITH THE STATE OF THE ART

In this section, we present a comparison of our proposed ResidualViT against the state-of-the-art methods for the NLTVG task. Our evaluation spans three benchmarks, considering both short and long video datasets. We assess the performance of ResidualViT in terms of grounding accuracy and computational efficiency, demonstrating its effectiveness. We also explore ResidualViT's generalization capabilities by applying it to the complementary task of Automatic Audio Description generation (Han et al., 2023).

| | R@1 | | | Avg. Cost Feature/sec |
|---|---|---|---|---|
| | IoU=0.1 | IoU=0.3 | IoU=0.5 | (GFLOPs) |
| CLIP (B/32) (Soldan et al., 2022) | 6.6 | 3.1 | 1.4 | 21.8 |
| ResidualViT (B/32) (ours) | 8.6 | 5.4 | 3.1 | 10.2 |
| ResidualViT (B/16) (ours) | 10.1 | 6.4 | 3.7 | 37.3 |
| ResidualViT (L/14) (ours) | 10.7 | 7.3 | 4.3 | 171.0 |

Table 3: **Long-form video state-of-the-art comparison.** ResidualViT outperforms the previous art both in accuracy and computational cost on the challenging long-form MAD dataset. In these experiments, ResidualViT was configured with $N=2$, a token dropping probability $p=85\%$, and the center token dropping strategy.

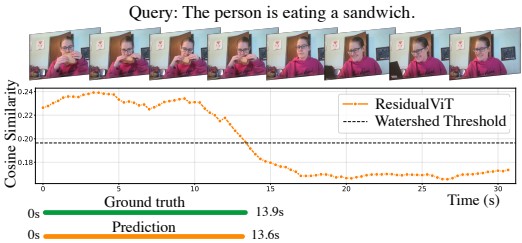

Query: The person is eating a sandwich.

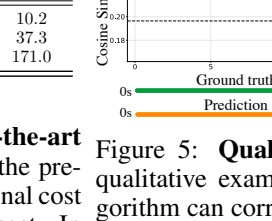

Figure 5: **Qualitative results.** We present a qualitative example in which our zero-shot algorithm can correctly ground the sentence in the video. We showcase the comparison between the ground truth annotation (**green**) and our top$-1$ prediction (**orange**).

**Natural Language Temporal Video Grounding (NLTVG) in Short Videos.** Table 2 summarizes the comparison against state-of-the-art grounding methods covering fully supervised, weakly supervised, pseudo-supervised, and zero-shot techniques for the short video setup. Our approach is directly comparable to the zero-shot methods (CLIP (Radford et al., 2019), UniVTG (Lin et al., 2023), MR-FVLM (Luo et al., 2024)) that do not train on downstream task data. Note that, in contrast to our approach, methods labeled as "Pseudo" use additional supervision for the temporal grounding task. While they do not use the existing annotations from the benchmarks' training sets, they employ readily available detectors for objects, actions, and scenes to assemble pseudo-sentences for annotating temporal locations in the downstream task training video datasets. These pseudo-sentences, together with their corresponding temporal locations, are used to train a grounding model in a fully supervised manner. See Section 2 for more details. Differently from these works, we do not use the training sets of the downstream benchmarks and, therefore, our model has not seen any of the downstream task data, which is a set-up that often happens in practical scenarios.

For each method, we report the grounding accuracy on the Charades-STA (Gao et al., 2017) and ActivityNet-Captions (Krishna et al., 2017) datasets, along with the average embedding cost per second. Previous methods have employed visual backbones like ResNet152, C3D, BLIP, VGG-19, and I3D (Carreira & Zisserman, 2017; He et al., 2016; Li et al., 2022; Simonyan & Zisserman, 2014; Tran et al., 2015) with respective costs of 11.6, 38.5, 55.5, 143.7, and 148.4 GFLOPs per feature. Table 2 also reports the grounding accuracy using the vanilla CLIP and our ResidualViT features across different backbones. For ResidualViT, we use motion-based token reduction with probability $p = 85\%$ and set the interleave parameter to $N = 2$. Our ResidualViT closely matches CLIP's grounding accuracy while reducing frame encoding costs by approximately $56\%$ across all ViT backbones. Particularly, on the Charades-STA dataset with the ViT-B/16 backbone, our method exhibits a negligible accuracy decrease compared to the standard CLIP encoding, whereas, for ViT-B/32 and L/14, we observe minor drops in the accuracy of $1\% - 1.5\%$. For the ActivityNet-Captions dataset, our method is on par with the directly comparable CLIP methods and achieves significant computational cost reduction; the accuracy decrease is less than $1\%$ across all configurations. When comparing against the two existing zero-shot methods, we find that UniVTG (Lin et al., 2023) significantly underperforms across all metrics compared to our results. In contrast, MR-FVLM (Luo et al., 2024) achieves comparable accuracy to our model, particularly at IoU=0.5 on the Charades-STA dataset, but at a substantially higher computational cost of 1337 GFLOPs per feature, compared to our 102.6 GFLOPs. This high cost in MR-FVLM is due to its use of the C3D backbone and the InternVideo-MM-L-14 model (Wang et al., 2022b).

Lastly, even though *our approach has not trained on the downstream task data*, our accuracy is, nevertheless, competitive against the previous art that has trained on both datasets. For the Charades-STA dataset, our approach achieves the best cost vs. accuracy trade-off over all the pseudo-supervised methods. For the ActivityNet-Captions dataset, the accuracy of our method with the B/16 backbone is on par or higher than 3 of the pseudo-supervised methods at a lower computational cost.

**Natural Language Temporal Video Grounding (NLTVG) in Long Videos.** In Table 3, we present additional results on the challenging long-form video MAD dataset (Soldan et al., 2022), contrasting our ResidualViT against the only zero-shot grounding baseline available, which is described in (Soldan et al., 2022). This existing zero-shot grounding algorithm employs a proposal-based approach, utilizing a multi-scale sliding window technique to generate potential video segment propos-

als. For each proposal, a single feature representation is computed by average pooling frame features whose temporal locations fall within the proposal's span. Finally, cosine similarity is computed between each sentence feature representation and each proposal feature representation. In contrast, our grounding algorithm (Appendix C) requires the number of similarity computations to be equal to the number of encoded frames, which significantly reduces the computational complexity. Specifically, the proposal-based method demands approximately $20\times$ more similarity computations compared to our approach.

Our results demonstrate that the grounding algorithm combined with ResidualViT visual features significantly outperforms the existing state-of-the-art. When using the same backbone (ViT B/32), our approach achieves relative improvements ranging from $43\%$ at IoU=0.1 to $128\%$ at IoU=0.5, while also being $53\%$ more efficient in feature extraction and requiring one order of magnitude fewer similarity computations. Additionally, accuracy consistently increases with the use of more computationally demanding backbones. For example, using the ViT B/16 backbone, our method achieves a $160\%$ increase in accuracy at IoU=0.5, despite a $73\%$ higher feature extraction cost compared to the baseline (Soldan et al., 2022). These findings highlight an excellent tradeoff between computational cost and improved accuracy. Additional comparisons and metrics can be found in Appendix D.

**Automatic Audio Description Task.** We benchmark the generalization capabilities of our ResidualViT by employing its feature representation on an additional downstream task related to long video understanding: Automatic Audio Description (Han et al., 2023). This task is akin to dense video captioning and aims to generate textual descriptions of relevant video moments, detailing the events and characters involved.

The approach for this task proposed in (Han et al., 2023) leverages two large-scale pre-trained models, such as CLIP (Radford et al., 2021), for visual feature extraction, and GPT-2 (Radford et al., 2019), for textual caption generation, connecting the two via a learned transformer encoder that aligns the visual and language features. To evaluate the quality of the ResidualViT visual representations, we replace the default CLIP visual features with ResidualViT features and perform the inference without any model fine-tuning. The evaluation is performed on a subset of the MAD dataset, specifically *MAD-eval-Named* (more details can be found in Section 4 of the AutoAD manuscript (Han et al., 2023)). Following the AutoAD setup, we extract visual features at five frames per second using the ViT-B/32 backbone. For ResidualViT, we set $N = 2$ and $p = 85\%$, achieving a $53\%$ reduction in frame encoding cost compared to CLIP. Moreover, to isolate the contribution of the visual features to the task solution, we evaluate the performance when no context audio descriptions or context subtitles are provided. No pretraining data is used, and the visual temporal context is set to 8 frames.

Under these conditions, the original CLIP features achieved a CIDEr score of 7.5, while our ResidualViT features resulted in a CIDEr score of 7.2. This experiment suggests that ResidualViT offers an excellent cost-performance tradeoff, with only a marginal performance reduction compared to the upper-bound performance of CLIP while significantly reducing the feature encoding cost.

Note that this experiment was conducted using a pre-trained model provided by the authors of (Han et al., 2023). This provided model differs from the one evaluated in the original AutoAD manuscript, so the performance results do not exactly match those reported in (Han et al., 2023). Nonetheless, this result provides evidence that our ResidualViT's visual representations are applicable to another video understanding task, video captioning, which is complementary to natural language video grounding, showcasing our model's flexibility and generalization capabilities.

### 4.3 QUALITATIVE RESULTS

Figure 5 presents a qualitative example of temporal grounding from the Charades-STA dataset, where our zero-shot grounding algorithm accurately predicts the temporal span corresponding to the textual query, "the person is eating a sandwich". This prediction is driven by the cosine similarity profile between the visual and language features, along with the watershed threshold, as illustrated in the figure. See Appendix C for details on the watershed threshold and the overall grounding algorithm. Appendix J shows additional visualizations and examples of failure cases.

## 5 CONCLUSION

We have developed a new approach for the efficient computation of transformer-based video features, exploiting temporal redundancy in videos via learnable temporal residual connections. The proposed approach is lightweight as it trains only a small number of parameters in the residual module while keeping the encoder fixed and does not require any additional training data as it is trained via distillation from existing (but costly) video encoders. We have demonstrated the benefits of the proposed approach on the natural language grounding task showing a significant reduction (up to 60%) in compute cost with marginal accuracy reduction. We believe that our work opens up the possibility of extending the distillation objective to incorporate richer interactions between visual and language representations, as well as exploring additional large-scale pre-trained models that natively model temporal relationships.

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

# APPENDIX

We provide the following additional information:

- **Token Dropping Strategies:** We present the different token dropping strategies that can be adopted in the token reduction module in Appendix A.

- **Motion-based Token-dropping Strategy:** Insights into the motion-based token-dropping strategy are provided in Appendix B, explaining the additional RAM requirements and the pre-processing of raw motion vectors.

- **Zero-shot Grounding Algorithm:** A thorough explanation of the implementation details of the zero-shot temporal grounding algorithm is presented in Appendix C.

- **Additional Comparison for Long-Form NLTVG** Additional analysis on the MAD dataset are available in Appendix D.

- **Supplementary Ablations:** In Appendix E, we conduct additional ablations on ResidualViT, exploring different token reductions strategies as presented in Appendix A and discussing the role of token-dropping probability. Additionally, we investigate two distinct strategies for computational savings: token merging and reduction of the spatial resolution of the input frames. We ablated the design of the distillation approach and showcased how different distillation objectives can achieve competitive performance. We conclude by ablating the interleave factor during distillation training.

- **Video Encoding Latency:** Appendix F empirically validates the wall-clock timings of ResidualViT, demonstrating significant time savings compared to a standard ViT model, despite requiring two forward passes.

- **Evaluation Metrics:** Appendix G details the metrics used to assess performance.

- **Implementation Details:** Appendix H provides useful implementation details.

- **Limitations:** In Appendix I we discuss the inherent limitations of our solution.

- **Qualitative Results:** We conclude with a showcase of several qualitative results in Appendix J, highlighting the practical effectiveness of our approach.

- **Feature Comparison under Full Supervision Setup:** As an additional test of the quality of our ResidualViT features, we investigate the performance of CG-DETR (Moon et al., 2023) when replacing the original CLIP features with our ResidualViT ones in Appendix K.

- **Additional task - Action Recognition:** In this Supplementary experiment, we test the performance of CLIP features against ResidualViT features on the task of action recognition on the Kinetics 400 dataset. Results are reported in Appendix L.

- **Additional task - Temporal Activity Localization:** In this Supplementary experiment, we test the performance of CLIP features against ResidualViT features on the task of temporal activity localization on the ActivityNet dataset. Results are reported in Appendix M.

## APPENDIX A   TOKEN DROPPING STRATEGIES

In Section 3.1, we introduced the ResidualViT architecture, which consists of the token reduction module ($\mathcal{R}$), the residual tokenizer ($\mathcal{A}$), and the transformer encoder ($\mathcal{E}_{\mathcal{V}}$). Here, we explore four practical implementations of the token reduction module when adopting the token dropping strategy (Ding et al., 2023; Haurum et al., 2023; Hou et al., 2022a; Liu et al., 2023b).

For a given frame $x_t$, which is transformed into a set of tokens $\mathcal{T}$, each strategy retains $(1 - p) \times |\mathcal{T}|$ tokens, where $p$ is the token reduction probability. Figure 6 visually depicts the four token reduction approaches we investigate. (i) The **random** strategy randomly samples tokens from the set. Conversely, (ii) the **uniform** strategy selects tokens from patches that are evenly distributed across the 2D grid of image patches, ensuring that the selected patches are spaced at regular intervals throughout the frame. (iii) The **center** strategy is designed to retain tokens of patches from the center of the frame. This strategy takes into consideration that, when shooting a video, we tend to center the frame around the subject or action being recorded. Finally, we design a data-dependent (iv)

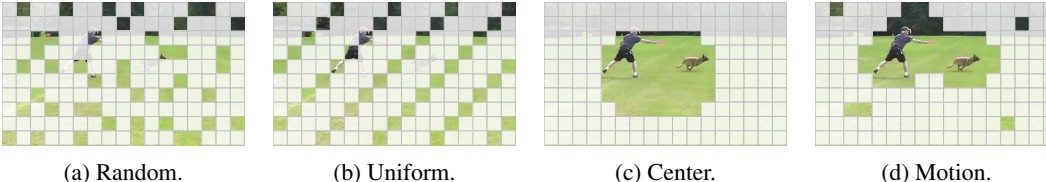

| (a) Random. | (b) Uniform. | (c) Center. | (d) Motion. |

Figure 6: **Token reduction strategies.** We implement three data-independent token reduction strategies (a-c) and one data-dependent one (d).

**motion** strategy. This strategy further exploits the characteristics of video data, which describes how characters, objects, and scenes evolve in time. We argue that motion is a valuable source of information readily accessible from encoded video files, providing information on which parts of the frame at time step $t + k$ differ from the frame at time step $t$. Consequently, we discard tokens representing patches with minimal motion, assuming their change relative to previous frames is negligible, and their information can be recovered through the residual token. See Appendix B for additional details about motion preprocessing and memory overhead.

## APPENDIX B IMPLEMENTATION DETAILS FOR MOTION-BASED TOKEN REDUCTION STRATEGY

Motion is a valuable and readily available source of information for determining which spatial regions of a frame have changed with respect to the previous one. To harness this information, our method employs a compressed video reader (AcherStyx, 2020) that extracts motion vectors directly from compressed video streams. Nevertheless, it is important to acknowledge that motion vectors extracted from raw video data typically exhibit a moderate level of noise, attributable to the inherent sparsity and optimization mechanisms of standard video compression techniques. To counteract this effect and derive a more reliable motion estimation, we compute the average motion across a short temporal window surrounding a target frame $x_t$. Specifically, we construct a set of motion vectors $M_v = \{m_i\}_{i=t-W_M/2}^{t+W_M/2}$, where each vector $m_i \in \mathbb{R}^{H' \times W' \times C'}$ corresponds to the motion information of frame $i$. Here, $H' = H/4$ and $W' = W/4$ are the reduced height and width dimensions, respectively, and $C' = 4$ signifies the channels in the motion vector, capturing the $(\Delta x, \Delta y)$ displacement of pixels with respect to adjacent frames (previous and following ones). The parameter $W_M$ denotes the size of the temporal window over which the motion is aggregated. As we are interested in the magnitude of the motion and not its direction, we compute the average $L_1$ norm along dimension $C'$ in the window $W_M$. Note that, at the start of the video ($t < W_M$) and at the end ($t > T - W_M$, where $T$ is the timestamp of the last frame), the window is reduced so that only the available motion vectors are aggregated, avoiding the need for padding.

We then upsample the computed motion magnitudes to the frame resolution $(H, W)$ and select the $1 - p$ frame tokens with the highest motion magnitudes at their patch's spatial location.

In our implementation, we set the motion window size $W_M = 11$. This setting implies that we incur an additional RAM memory consumption that is proportional to the cost of storing a frame in memory. We can estimate the memory cost as follows. The memory consumption of a frame can be expressed as $M_F = H \times W \times 3$, while for motion vectors $M_{Mv} = (H/4) \times (W/4) \times 4 \times M_v$, resulting in a total memory cost $M_F + M_{Mv}/M_F = \sim 1.9\times$. Note that this memory overhead does not affect GPU memory availability as the motion vectors are not required to be moved to such a device for processing. To determine the value of $W_M$,

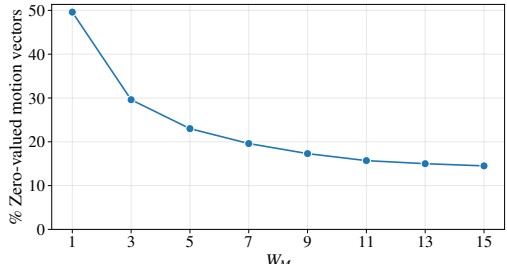

Figure 7: **Optimal $W_M$ Hyperparameter Value.** The plot shows the average percentage of zero-valued motion vectors on the Charades-STA dataset as the aggregation window size $W_M$ varies. The trend flattens beyond $W_M = 11$, indicating diminishing returns. Therefore, we choose $W_M = 11$ as our default parameter.

we measure the average percentage of zero-value motion vectors in the Charades-STA dataset. As shown in Figure 7, we find that for $W_M = 1$, roughly $50\%$ of motion vectors are zero while considering $W_M = 11$ reduces this value to less than $15\%$. We do not observe a significant reduction beyond $W_M = 11$. For simplicity, we keep this parameter constant across datasets.

Finally, note that we utilize motion information to identify frame patches that have likely undergone significant transformations relative to preceding frames. This strategy enables us to provide the transformer encoder of ResidualViT ($\mathcal{E}_S$) with patches expected to exhibit less redundancy with previous frames. Importantly, this approach does not supply the encoder with motion information, meaning the network remains unaware of explicit motion patterns.

## APPENDIX C  ZERO-SHOT GROUNDING ALGORITHM

As discussed in Section 3, this work focuses on the task of natural language video grounding, which involves temporally localizing a natural language description within a single video. Given the fine-grained temporal localization requirements of the task, dense frame sampling and encoding are indispensable, making it an ideal testbed for our efficient ResidualViT approach. This section details feature encoding and describes the motivations for addressing the task in a zero-shot setting.

We argue that the zero-shot setting holds valuable properties. Firstly, algorithms evaluated in a zero-shot manner are not prone to be affected by the inherent biases of the downstream datasets, which have shown to be a danger for this task (Otani et al., 2020; Soldan et al., 2022; Zhang et al., 2021). Additionally, models exhibiting strong zero-shot capabilities typically demonstrate enhanced generalization to unseen datasets, thereby increasing their versatility and utility. Secondly, from a practical standpoint, relying on multiple specialized models for each new dataset can severely limit the scalability and versatility of systems. In contrast, a unified model that excels in zero-shot settings streamlines system architecture and boosts scalability and adaptability. Such models simplify the maintenance and deployment of deep learning applications and readily adjust to new challenges without the need for extensive retraining. Third, the zero-shot approach promotes environmental sustainability. This approach significantly curtails the computational demands by drastically reducing the necessity for ongoing retraining on possibly extensive datasets, thus lowering energy consumption and the associated carbon footprint. Employing large pre-trained models in a zero-shot manner optimizes their efficacy while minimizing further environmental impacts. We strive to pursue zero-shot evaluation in this work for all these reasons.

**Visual Encoding.** Our algorithm begins with encoding a set of video frames $\mathcal{X} = \{x_t\}_{t=1}^{n_v}$ through a designated visual encoder (either a standard ViT or our ResidualViT). This process generates a series of frame features $\{f_t\}_{t=1}^{n_v}$. When employing ResidualViT, in line with the approach illustrated in Figure 2a, we utilize a sliding window mechanism that concurrently processes $N + 1$ frames. The first frame in each window is encoded by the foundation model encoder $\mathcal{E}_V$, with the resulting features stored for subsequent use. The following $N$ frames are processed by encoder $\mathcal{E}_S$ (Figure 2b), which takes as input the frame tokens and the cached feature of the first frame of the window. The residual feature is first transformed into the residual token via the residual tokenizer. Subsequently, in the reduction module, frame tokens are reduced according to a particular strategy and token reduction probability $p$. Finally, these sparse visual tokens are concatenated with the residual token and forwarded to the visual encoder $\mathcal{E}_V$.

**Language Encoding.** The language encoder is kept frozen throughout our experiments and initialized with CLIP weights corresponding to the specific version of the visual encoder (ViT-B/32, ViT-B/16, or ViT-L/14). To solve the task, each sentence $s$ is first tokenized and then processed through the language encoder to derive a single sentence feature $g_l$.

**Grounding Algorithm.** For the grounding task, cosine similarity between each frame embedding and the sentence embedding is calculated, creating a temporal sequence of similarity scores $\{S_t\}_{t=1}^{n_v}$. We post-process the similarity profiles with a moving average smoothing operation with window size $W_{MA}$.

Finally, inspired by methods in prior work such as (Diwan et al., 2023; Lei et al., 2021a), we implement a watershed algorithm (Roerdink & Meijster, 2000) for moment prediction. In this step, group consecutive timesteps where the similarity scores exceed a given threshold, effectively delineating temporally contiguous segments. The start and end timesteps of these segments constitute our mo-

| Grounding Algorithm | Use Downstream Task Data | Features | Visual Backbone | R@1 | | | R@5 | | | Avg. Cost Feature/sec (GFLOPs) |
| --- | --- | --- | --- | --- | --- | --- | --- | --- | --- | --- |
| | | | | IoU=0.1 | IoU=0.3 | IoU=0.5 | IoU=0.1 | IoU=0.3 | IoU=0.5 | |
| DenoiseLoc (Xu et al., 2023) | ✓ | CLIP | ViT-B/32 | 1.1 | 0.9 | 0.5 | 4.1 | 3.3 | 2.2 | 21.8 |
| 2D-TAN (Zhang et al., 2020) | ✓ | CLIP | ViT-B/32 | 3.2 | 2.5 | 1.6 | 11.9 | 9.3 | 5.7 | 21.8 |
| Moment-DETR (Lei et al., 2021b) | ✓ | CLIP | ViT-B/32 | 3.6 | 2.8 | 1.7 | 13.0 | 9.9 | 5.6 | 21.8 |
| VLG-Net (Soldan et al., 2021) | ✓ | CLIP | ViT-B/32 | 3.6 | 2.8 | 1.7 | 11.7 | 9.3 | 6.0 | 21.8 |
| CONE (Hou et al., 2022b) | ✓ | CLIP | ViT-B/32 | 8.9 | 6.9 | 4.1 | 20.5 | 16.1 | 9.6 | 21.8 |
| SOONet (Pan et al., 2023) | ✓ | CLIP | ViT-B/32 | 11.3 | 9.0 | 5.3 | 23.2 | 19.6 | 13.1 | 21.8 |
| SnAG (Mu et al., 2024) | ✓ | CLIP | ViT-B/32 | 10.4 | 8.5 | 5.5 | 24.4 | 20.3 | 13.4 | 21.8 |
| RGNet (Hannan et al., 2023) | ✓ | CLIP | ViT-B/32 | 12.4 | 9.5 | 5.6 | 25.1 | 18.7 | 10.9 | 21.8 |
| Proposals (Soldan et al., 2022) | ✗ | CLIP | ViT-B/32 | 6.6 | 3.1 | 1.4 | 15.1 | 9.9 | 5.4 | 21.8 |
| Watershed | ✗ | CLIP | ViT-B/32 | 8.7 | 5.5 | 3.2 | 21.1 | 13.0 | 7.3 | 21.8 |
| Watershed (ours) | ✗ | ResidualViT | ViT-B/32 | 8.6 | 5.4 | 3.1 | 20.5 | 12.6 | 6.9 | $10.2_{(-53\%)}$ |
| Watershed | ✗ | CLIP | ViT-B/16 | 10.8 | 6.8 | 3.9 | 24.5 | 15.2 | 8.5 | 84.3 |
| Watershed (ours) | ✗ | ResidualViT | ViT-B/16 | 10.1 | 6.4 | 3.7 | 23.5 | 14.6 | 8.1 | $37.3_{(-56\%)}$ |
| Watershed | ✗ | CLIP | ViT-L/14 | 13.3 | 8.6 | 5.0 | 28.5 | 18.2 | 10.3 | 389.2 |
| Watershed (ours) | ✗ | ResidualViT | ViT-L/14 | 10.7 | 7.3 | 4.3 | 24.4 | 16.6 | 9.3 | $171.0_{(-56\%)}$ |

Table 4: **Long-form video state-of-the-art comparison.** ResidualViT outperforms the previous art both in accuracy and computational cost on the challenging long-form MAD dataset. In these experiments, ResidualViT was configured with $N=2$, a token dropping probability $p=85\%$, and the center token dropping strategy.

ment predictions. Multiple predictions are sorted based on the highest frame-sentence similarity in their span.

For short-video datasets, such as Charades-STA and ActivityNet-Captions, we compute the threshold as a scaled average of the scores, given by $\frac{\alpha}{n_v} \sum_{t=1}^{n_v} S_t$, where $\alpha$ is a scaling factor. Conversely, for the long-form MAD dataset, we normalize the scores to the range $[0, 1]$ and apply a fixed threshold $\beta$, an approach that mitigates the influence of low-relevance similarities in longer sequences that can otherwise skew the average similarity score. Appendix J presents several qualitative results showcasing the aforementioned similarity profile.

# APPENDIX D   ADDITIONAL COMPARISON IN LONG-FORM NLTVG

In this section, we present additional grounding results for the long-form MAD dataset. Table 4 builds on Table 3 from the main paper by incorporating results from supervised state-of-the-art methods and zero-shot watershed accuracy using CLIP features.

We begin by emphasizing that our zero-shot watershed-based grounding algorithm, detailed in Section Appendix C, significantly outperforms the proposal-based method introduced by Soldan et al. (2022). By comparing rows 9 and 10 of the table, where both algorithms utilize the same visual backbone (CLIP ViT-B/32), we isolate and evaluate their individual contributions. Our zero-shot watershed-based approach demonstrates superior accuracy, with relative improvements ranging from $43\%$ to $128\%$. Remarkably, our zero-shot results are comparable with, or even surpass, several fully supervised methods listed in rows 1 through 8.

Table 4 also enables a direct comparison of different backbone features while keeping the grounding algorithm fixed, thereby contrasting CLIP with our ResidualViT. For ResidualViT, we utilize configurations of $N = 2$, $p = 85\%$, and a center token dropping strategy, resulting in an embedding cost reduction of $53\%$ to $56\%$.

When using the ViT-B/32 backbone (rows 10-11), ResidualViT reduces computational costs by approximately $53\%$, with an average performance degradation of just $0.1\%$ compared to CLIP weights, a negligible decrease. Similarly, employing the ViT-B/16 backbone (rows 12-13) ResidualViT achieves a $56\%$ reduction in computation with respect to CLIP, accompanied by an average performance drop of $0.5\%$. For the larger ViT-L/14 backbone, the average performance drop is $1.5\%$, with the most significant decrease occurring for the less stringent metric (R@1 IoU=0.1). These results demonstrate that ResidualViT offers an excellent performance-to-cost reduction trade-off across all ViT variants within the MAD dataset.

| | Drop Strategy | Charades-STA R@1 | | Avg. Cost per Feature (GFLOPs) | Memory Cost per Feature (normalized) |
|---|---|---|---|---|---|
| | | IoU=0.5 | IoU=0.7 | | |
| ViT-L/14 | – | 42.9 | 24.1 | 233.4 | 1× |
| | Random | 40.8 | 23.3 | 102.6 | 1× |
| | Uniform | 39.6 | 22.5 | 102.6 | 1× |
| | Center | 38.6 | 21.1 | 102.6 | 1× |
| | Motion | 41.5 | 23.8 | 102.6 | 1.9× |

Table 5: **Token reduction strategy ablation for ResidualViT.** We ablate four different token reduction strategies on the Charades-STA dataset. For all, we fix the token reduction probability to 85%. Memory cost is normalized according to the baseline memory footprint.

| | Drop Strategy | Charades-STA R@1 | | Avg. Cost per Feature (GFLOPs) | Memory Cost per Feature (normalized) |
|---|---|---|---|---|---|
| | | IoU=0.5 | IoU=0.7 | | |
| ViT-L/14 | – | 42.9 | 24.1 | 233.4 | 1× |
| | Random | 20.8 | 9.5 | 102.6 | 1× |
| | Uniform | 21.0 | 10.6 | 102.6 | 1× |
| | Center | 25.8 | 13.2 | 102.6 | 1× |
| | Motion | 28.5 | 14.5 | 102.6 | 1.9× |

Table 6: **Token reduction strategy ablation for CLIP.** We ablate four different token reduction strategies on the Charades-STA dataset. For all, we fix the token reduction probability to 85%. Memory cost is normalized according to the baseline memory footprint.

## APPENDIX E    ADDITIONAL ABLATIONS

In this section, we delve deeper into the design choices of ResidualViT by performing ablation studies on its token reduction mechanisms and distillation strategy. We begin by testing several designs for token-dropping strategies as presented in Appendix A and discussing the role of token-dropping probability. Next, we explore an alternative approach to the token reduction module by replacing token-dropping with a token merging strategy (Bolya et al., 2022). We then assess the impact of reducing input frame resolution on the total number of tokens, providing insights into its effectiveness as a computational saving technique. Finally, we investigate an alternative distillation objective that eliminates the need for language annotations.

Note that, while semantically aware token reduction strategies (Ding et al., 2023) could be incorporated, we leave this for future work due to their additional computational demands (*i.e.*, complex token relevance computation at each level of the transformer encoder).

**Token Reduction Module Ablation - Token Drop Strategy.** Here, we ablate the different token reduction strategies presented in Appendix A. In Table 5, we contrast the grounding accuracy of the CLIP model (first row) against our ResidualViT encoder. The lowest grounding accuracy is achieved by the center token reduction strategy with relative drops (vs. the CLIP model, first row) in the range of $10-12\%$. Uniform sampling produces slightly better accuracy with relative drops in the range of $6-7\%$. The second-best performing method is random, which decreases the drop to $3-5\%$. Finally, the motion-based strategy closely matches the grounding accuracy of the CLIP baseline with a relative drop in the range of $1-3\%$. Given the fixed token reduction probability, all settings result in a cost reduction of $56\%$ with respect to the naive CLIP frame encoding baseline.

Additionally, in Table 6, we report the accuracy when the different token reduction strategies are applied to the CLIP model. In this case, we observe much wider differences between different token reduction strategies, where random and uniform strategies perform the worst with a relative accuracy drop in the range of $50-60\%$. The center token reduction strategy provides better accuracy, reducing the losses to $40-45\%$, while motion provides the best trade-off with a $34-40\%$ drop.

It is important to observe that our model (Table 5) provides a certain level of resilience to the type of token reduction strategy compared to the baseline CLIP model (Table 6). This finding suggests that for ResidualViT, token reduction strategies that avoid motion computation can serve as viable alternatives, especially in scenarios with limited memory or restricted computational resources. We attribute this finding to the learnable temporal residual connection, which enables the model to effectively compensate for the discarded tokens.

**Token reduction probability.** Here, we assess how varying the token reduction probability affects the performance of both the baseline CLIP

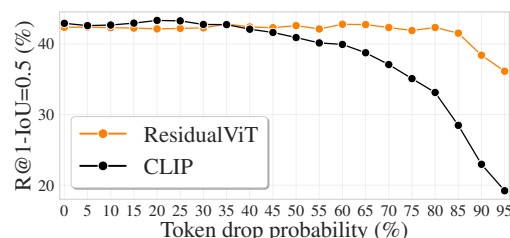

Figure 8: **Token drop probability ablation**. We showcase the performance of CLIP (**black**) and our ResidualViT (**orange**) when progressively increasing the token drop probability.

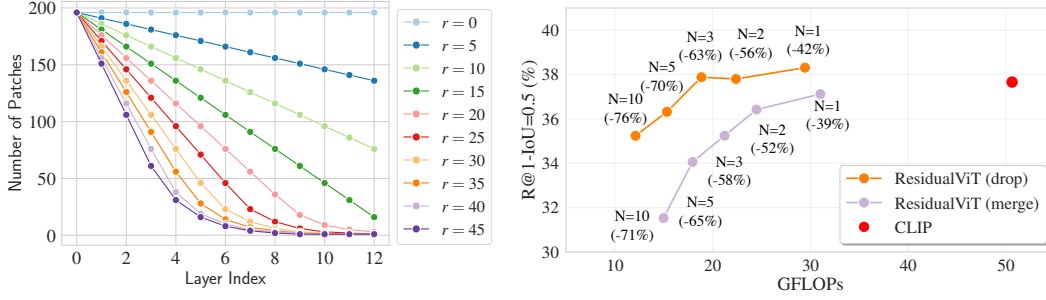

(a) Tokens decay profile for different $r$ factors.

(b) Drop vs Merge ablation.

Figure 9: **Token dropping vs merging**. (a) We illustrate the relationship between the ViT layer index and the number of tokens resulting from the token merging operation for several canonical merging factors (r). (b) We compare the cost (GFLOPs) vs performance (R@1-IoU=0.5) for CLIP and ResidualViT. We present CLIP without any token reduction strategy (**red**), against our ResidualViT when the token reduction is token dropping (**orange**) or token merging (**purple**). The ablation can conclude that token merging is less favourable due to lower performance at a comparable cost reduction.

model and our ResidualViT model. As depicted in Figure 8, the CLIP model (**black**) demonstrates a degree of robustness to the dropped tokens, maintaining relatively stable grounding accuracy until the token reduction probability reaches $35 - 40\%$. Beyond this setting, we observe a gradual decline in accuracy, which becomes more pronounced when the probability exceeds $80\%$. In contrast, thanks to our model design, ResidualViT (**orange**) exhibits a higher tolerance to dropped tokens, retaining relatively high grounding accuracy up to $p{=}85\%$ of dropped tokens.

**Token Reduction Module Ablation - Token Merging.** Our ResidualViT is agnostic to the implementation of the token reduction method. Therefore, we ablate replacing the token dropping strategy (Liu et al., 2023b; Hou et al., 2022a) with token merging (Bolya et al., 2022), which has shown promising results in reducing the inference time of pre-trained ViT models.

This solution opts for merging a fixed number of tokens per layer, denoted by the $r$ parameter. Within each transformer block, the set of frame tokens at layer $l$, denoted as $\mathcal{T}^l$, is divided into two subsets: $\mathcal{T}^l_{\mathbf{odd}}$, containing tokens at odd indices, and $\mathcal{T}^l_{\mathbf{even}}$, containing tokens at even indices. A bipartite matching is computed over the two sets by calculating the cosine similarity between the *key* embeddings of tokens derived from the self-attention mechanism. The $r$ edges of the bipartite graph characterized by the highest similarity define the assignment. The connected tokens are then merged together via a weighted sum, where each token weight represents how many tokens were previously aggregated in it. Note that neither the `[CLS]` token nor the residual token is merged with the frame tokens. Following the bipartite assignment, the maximum number of token mergers per layer is limited to half of the total number of tokens available at layer $l$ ($min(|\mathcal{T}^l|/2, r)$).

This token-reduction strategy has the potential to reduce the information loss that affects the token dropping strategy, as the content of the tokens is retained even if their number is reduced. However, it presents other limitations. (i) Due to the progressive nature of the merging operation (after each transformer layer), to achieve a comparable cost reduction to token dropping, the $r$ parameter must be large. (ii) When the $r$ factor is moderately large, the majority of the tokens are merged together. This effect is showcased in Figure 9a, where we see that for higher values of $r$, the number of tokens reduces to one quite early in the network (*e.g.*, around the depth of layer 8 for $r = 45$).

In Figure 9b, we conduct a comparative analysis of the token dropping and token merging strategies. For both strategies, we employ the ViT-B/16 backbone model. We set $p = 85\%$ and used the motion-based strategy for token dropping. We set $r = 40$ for token merging. We report R@1-IoU=0.5 grounding accuracy on Charades-STA. We compare the CLIP baseline in **red** against ResidualViT equipped with token dropping (**orange**) or token merging (**purple**).

Figure 9b shows token merging achieves overall lower grounding accuracy and incurs a significantly higher computation cost for its highest grounding accuracy setting (∼30 GFLOPs for token merging with N=1 versus ∼17 GFLOPs for token dropping with N=3). Nonetheless, both strategies are capable of effectively reducing the cost with respect to the CLIP baseline (**red**). This result validates our design choices for the token reduction module of our ResidualViT.

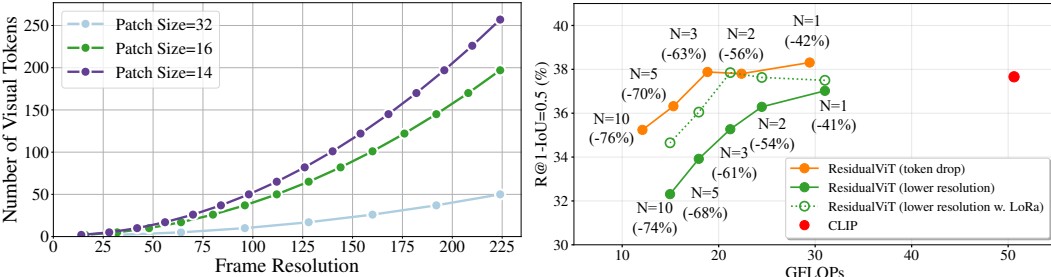

(a) Number of visual tokens vs. frame resolution. (b) Token Drop vs. lower resolution input ablation.

Figure 10: **Token drop vs lower resolution**. (a) We illustrate the relationship between frame resolution and number of tokens as a function of three canonical patch sizes. (b) We compare the cost (GFLOPs) vs performance (R@1-IoU=0.5) for CLIP and ResidualViT. We present CLIP without any token reduction strategy (**red**), against our ResidualViT with token drop (**orange**) or with lower input resolution (**green**). For the lower resolution setting, we additionally explore using LoRa (Hu et al., 2021) adapters to finetune the input 2D convolution that implements the patchyfication operation.

**Spatial Resolution Ablation.** An alternative to directly manipulating the number of frame tokens involves adjusting the spatial resolution of input frames. This strategy has proven effective in dual-branch architectures (Feichtenhofer et al., 2019), where one branch processes a few high-resolution frames, and the other handles many low-resolution frames. In this section, we compare our approach against this strategy.

In particular, we forward the full resolution I-frame to the ViT encoder $\mathcal{E}_\mathcal{V}$ and $N$ low-resolution P-frames to the ResidualViT encoder $\mathcal{E}_\mathcal{S}$. The number of tokens $|\mathcal{T}|$ for an input frame with resolution $(H, W)$ is calculated as $|\mathcal{T}| = \frac{H \times W}{P^2}$, where $P$ denotes the patch size. Consequently, reducing the frame resolution directly decreases the total number of tokens produced from the frame. We provide the relationship between frame resolution, patch size, and number of tokens in Figure 10a, examining trends across three canonical patch sizes: $P \in \{14, 16, 32\}$.

Subsequently, in Figure 10b, we contrast the performance of two variations of ResidualViT. One variant employs token dropping (**orange**), while the other utilizes a reduced input frame resolution (**green**), both using the ViT-B/16 backbone model. For the token dropping variant, we set $p = 85\%$, and for reduced resolution, we adjust the spatial dimensions to $H = W = 96$ pixels (as opposed to the default $H = W = 224$). These modifications yield comparable reductions in computational cost, as demonstrated by the alignment of the data points along the x-axis. However, our results indicate that reducing the input resolution is less effective than employing token dropping in terms of performance (y-axis). Note that, in all experiments where the token reduction module is modified, we re-train the residual tokenizer to ensure consistent performance evaluation.

We hypothesize that reducing the input frame resolution compromises the quality of the token representations inputted to the transformer. The process of converting image frames into tokens is implemented through a 2D convolution where both the kernel size and stride are set to the patch size. Previous research has indicated that although convolutional kernels can generalize to different resolutions, substantial changes in resolution can negatively impact performance (Kannojia & Jaiswal, 2018; Richter et al., 2021). In our experiments, to match the computational cost reductions observed with the token reduction strategy, we decreased the resolution of inputs to the ResidualViT encoder by a factor of four. To address the resulting resolution mismatch, we explored fine-tuning the 2D convolutional layers using LoRa adapters (Hu et al., 2021). This adjustment helps account for the impact of lower-resolution inputs on token representation quality. Our findings show that incorporating LoRa adapters with lower-resolution inputs improves accuracy across all $N$ values and achieves accuracy comparable to the token drop strategy for $N = 3$. However, the token drop strategy consistently outperforms this approach while maintaining the advantage of not requiring any weight modifications to the encoder $\mathcal{E}_\mathcal{V}$.

**Distillation strategy.** To evaluate our distillation approach, we replace the NCE loss (Equation 2) with a Mean Squared Error (MSE) loss computed between the frame features $||f^{\mathcal{S}}_{i,t+k} - f^{\mathcal{V}}_{i,t+k}||_2$. This alternative setup, illustrated in Figure 11a, eliminates the need for language annotations. Figure 11b presents the results, showing that the MSE loss performs competitively with the NCE loss. Both strategies demonstrate nearly identical performance, suggesting that our distillation method's

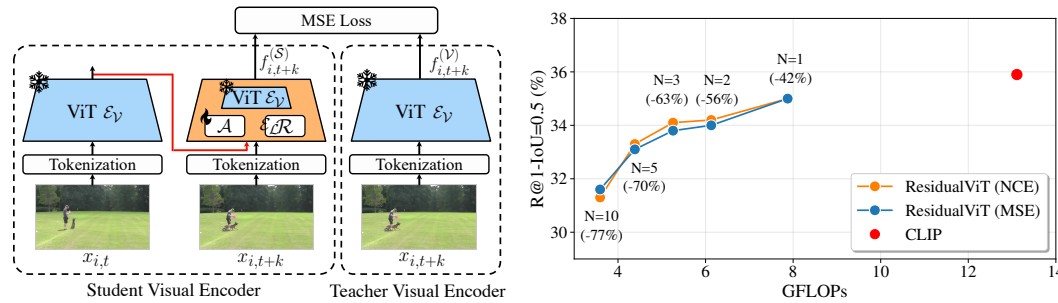

(a) Distillation pipeline.

(b) Downstream performance comparison.

Figure 11: **Distillation loss ablation.** We ablate replacing the NCE loss (Equation 2) with a Mean Square Error (MSE) loss. (a) Depicts the distillation pipeline when the MSE loss is used. (b) Summarizes the downstream performance comparison. The **red** represents CLIP's performance, while the **orange** and **blue** curves represent the performance of ResidualViT on the Charade-STA dataset when the distillation uses the original NCE loss or the MSE loss respectively. We perform this ablation adopting the ViT-B/32 backbone. We conclude that the MSE loss, which does not require language annotations, produces near-identical results.

effectiveness is robust to the choice of loss function. This ablation was conducted using the ViT-B/32 backbone, with all training and testing hyperparameters held constant.

**Frame Rate Ablation.** In this section, we evaluate the performance-cost trade-off between frame rate and computational cost for CLIP and ResidualViT on the Charades-STA dataset.

Figure 12 illustrates the performance of both models on the NLTVG task as the frame rate varies from 0.5 to 3.0 (our default value). At the default frame rate of 3.0, CLIP achieves an R@1-IoU=0.5 score of 35.9, while ResidualViT achieves 34.2—a slight performance drop, but with an approximate 53% reduction in encoding cost. As the frame rate decreases, both methods exhibit a steady decline in accuracy. However, it is noteworthy that ResidualViT at FPS=3 incurs a lower cost than CLIP at FPS=2 while achieving comparable accuracy.

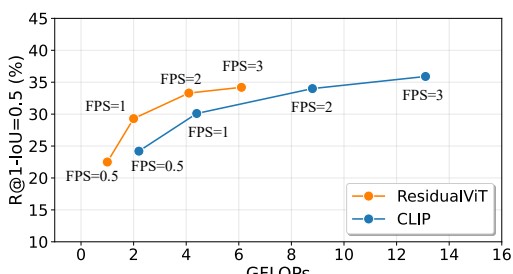

Figure 12: **Frame Rate Ablation.** We compare CLIP (**blue**) and ResidualViT (**orange**) features on the Chardes-STA dataset for varying frame rates. The figure presents the accuracy (y-axis) vs. cost (x-axis) trade-off.

Additionally, ResidualViT at FPS=2 outperforms CLIP at FPS=1, with similar computational cost.

Finally, we observe that the decrease in accuracy for ResidualViT as the FPS decreases becomes steeper than for CLIP. We believe that this is due to the large temporal gap between consecutive frames, which hinders the ability of the residual tokenizer to provide valuable information when computing P-features.

**Training Interleave Factor ($N_{\text{Train}}$) Ablation.** In this section, we evaluate how varying the interleave factor ($N_{\text{Train}}$) during training impacts ResidualViT's accuracy and computational cost for different $N$ values during inference. Additionally, we explore whether different frame sampling strategies during training affect the model's final accuracy. We consider two distinct sampling approaches: (a) Sample $N_{\text{Train}}$ frames per training video at a constant frame rate. (b) Extract $N_{\text{Train}}$ frames at a constant frame rate, but randomly subsample the frames before inputting them into the network. Our findings are summarized in Figure 13, where we present the accuracy vs. cost trade-off on the Charades-STA dataset using the B/32 backbone.

In this experiment, we train ResidualViT with varying $N_{\text{Train}} \in 3, 5, 10$ and test these models with $N \in 1, 2, 3, 5, 10$. Note that $N_{\text{Train}} = 3$ is our default setting, used for all other results in the manuscript.

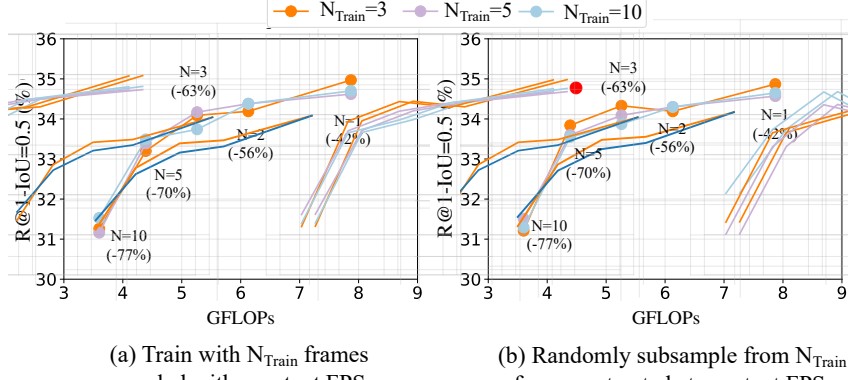

(a) Train with $N_{Train}$ frames sampled with constant FPS.

(b) Randomly subsample from $N_{Train}$ frames extracted at constant FPS.

Figure 13: **Training interleave factor ($N_{Train}$) ablation.** We compare the accuracy of ResidualViT when three different values of $N_{Train} \in \{3, 5, 10\}$ are used, and two different frame sampling strategies are implemented. In particular, we investigate (a) using all $N_{Train}$ frames sampled at a constant FPS$= 1.0$, and (b) sampling a random number of frames from the $N_{Train}$ frames extracted at a constant FPS$= 1.0$. Results are reported on the Chardes-STA dataset using the B/32 backbone.

Focusing on Figure 13(a), we observe that different values of $N_{Train}$ produce very similar results, with $N_{Train} = 10$ showing slightly better performance for $N = 5$ and $N = 10$ compared to models trained with $N_{Train} = 3$.

Figure 13(b) supports the same conclusion. In this case, no clear advantage is observed for larger $N_{Train}$, as accuracy remains very similar across all configurations. Interestingly, $N_{Train} = 3$ (our default setting) shows slightly better accuracy for $N = 1$, $N = 3$ and $N = 5$.

## APPENDIX F    RESIDUALVIT RUNTIME

In our manuscript, we have focused on characterizing the computational cost reductions in terms of GFLOPs. However, our system introduces a dependency where P-feature computation relies on the prior computation of I-features. Specifically, the I-features are first processed through the ViT encoder $\mathcal{E}_{\mathcal{V}}$, followed by the computation of P-features via the ResidualViT encoder $\mathcal{E}_{\mathcal{S}}$, which also incorporates the residual token. This design necessitates two sequential forward passes through distinct encoders, prompting us to examine the encoding latency costs inherent to this approach. One possible way to mitigate the latency due to this sequential dependency is via parallel processing via batching of the I-features, followed by batching of the P-features.

In Figure 14, we present the forward pass wall-clock latency as a function of batch size, comparing the timings for a standard ViT-L/14 model and our ResidualViT, which employs the

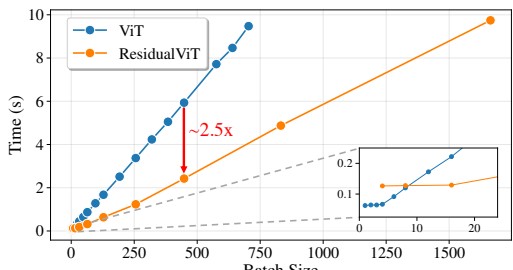

Figure 14: **Inference time comparison**. When varying the batch size, we showcase the runtime difference of a standard ViT (**blue**) against our ResidualViT (**orange**). We demonstrate that our approach is $\sim 2.5 \times$ faster than a standard ViT. Moreover, for the same time budget (*i.e.*, 10 seconds), we can accommodate $\sim 2.5 \times$ more samples in the batch without incurring Out Of Memory issues.

same ViT-L/14 backbone. For each batch size, the total time for ResidualViT is calculated as the sum of the time taken to compute the I-features and the time to process the P-features.

The graph indicates that our ResidualViT is more time-efficient than the ViT baseline, benefiting from our design optimized for efficient video encoding. In practice, our architecture requires roughly 2.5 times less wall-clock time to encode frames into features across most batch sizes. Additionally, when the encoding time is constrained, *e.g.* 10 seconds, the baseline model can process a batch size of approximately 700 frames, whereas ResidualViT can handle a batch size of about 1700 frames.

Note that, in the regime of small batch sizes (*i.e.*, $\leq 8$), highlighted in the zoomed box in Figure 14, the ViT model proves more economical compared to ResidualViT. Nonetheless, it is crucial to remember that our focus is on efficiently encoding numerous video frames for dense tasks, making ResidualViT the preferred choice under these conditions.

These experiments were performed using a single NVIDIA V100 GPU. Timings for each batch size were obtained by averaging results from 100 consecutive forward passes to ensure statistical reliability. To guarantee precise timing measurements, we employed the PyTorch function `torch.cuda.synchronize()`, which halts the execution of the code until all pending GPU operations are completed. This function is critical for avoiding discrepancies in timing due to asynchronous GPU execution.

## APPENDIX G   EVALUATION CRITERIA

In our experimental section, we measure performance via recall at rank $K$ for intersection over union (IoU) larger than $\theta$ (R@$K$-IoU=$\theta$). Given a ranked set of video moments, this metric measures if any of the top-$K$ ranked moments have an IoU larger than $\theta$ with the ground truth temporal endpoints. Following prior work (Gao et al., 2017; Hendricks et al., 2017), we report Recall@$K$ for IoU=$\theta$ with $K \in \{1, 5\}$ and $\theta \in \{0.5, 0.7\}$.

The computational cost for video encoding is quantified using Giga Floating Point Operations per Second (GFLOPs). This metric represents the average video encoding cost per second, calculated as the product of the computational cost to encode a single frame and the frame rate, which denotes the number of frames processed per second.

## APPENDIX H   IMPLEMENTATION DETAILS

We build on the publicly available OpenCLIP (Ilharco et al., 2021) implementation and use the default training parameters and loss with the exceptions noted next. Our method is trained on video-text pairs from the WebVid-2.5M dataset (Bain et al., 2021) for 5 epochs. Our batch size is 2048 for ViT-B/32 and ViT-B/16 models and 1536 for ViT-L/14. We encode one frame using the visual encoder $\mathcal{E}_\mathcal{V}$ and the three subsequent frames ($N_{\text{Train}} = 3$) with ResidualViT encoder $\mathcal{E}_\mathcal{S}$. For all experiments, we use a constant learning rate of $0.0005$ while weight decay and warmup are disabled. All model training is performed on 4 V100 GPUs, while inference only requires 1 V100 GPU. For the grounding algorithm, we set $W_{MA} = 15$ and $\alpha = 1.0$ for Charades-STA, $W_{MA} = 15$ and $\alpha = 0.95$ for ActivityNet-Captions and $W_{MA} = 7$ and $\beta = 0.7$ for MAD. At inference time, videos are processed at 3 frames per second for Charades-STA, 1 frame per second for ActivityNet-Captions, and 5 frame per second for MAD. GFLOPs are measured via the fvcore library (FAIR, 2020).

## APPENDIX I   LIMITATIONS

We acknowledge several technical limitations of our approach. Firstly, our method is specifically designed for the Vision Transformer (ViT) architecture, making it less applicable to other architectures, such as convolutional or recurrent neural networks. Nonetheless, we argue that transformer-based models have proven to be among the most versatile and scalable options in the deep learning landscape, supporting their continued adoption and adaptation.

Secondly, ResidualViT is optimized for dense video processing tasks, which may limit its efficacy in scenarios that benefit from sparse frame sampling, such as action recognition or video retrieval. For such applications, the semantic continuity captured by the residual token across temporally distant frames may not be sufficient, suggesting a potential area for future research.

Thirdly, our solution's effectiveness heavily relies on the quality of the underlying large pre-trained foundation model, such as CLIP (Radford et al., 2021). Consequently, any inherent biases or limitations in the pre-trained model's weights could adversely affect our method's performance.

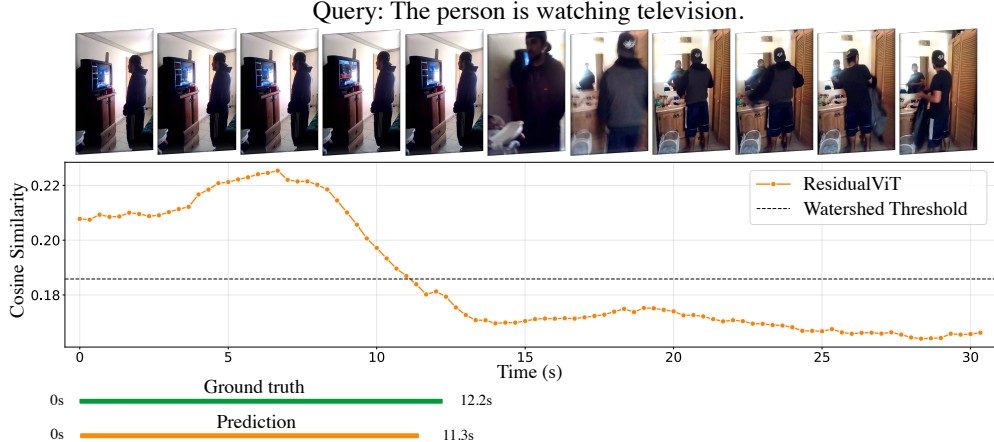

(a) **Grounding example.** We observe an IoU $= 0.93$ between the ground truth moment and the predicted one.

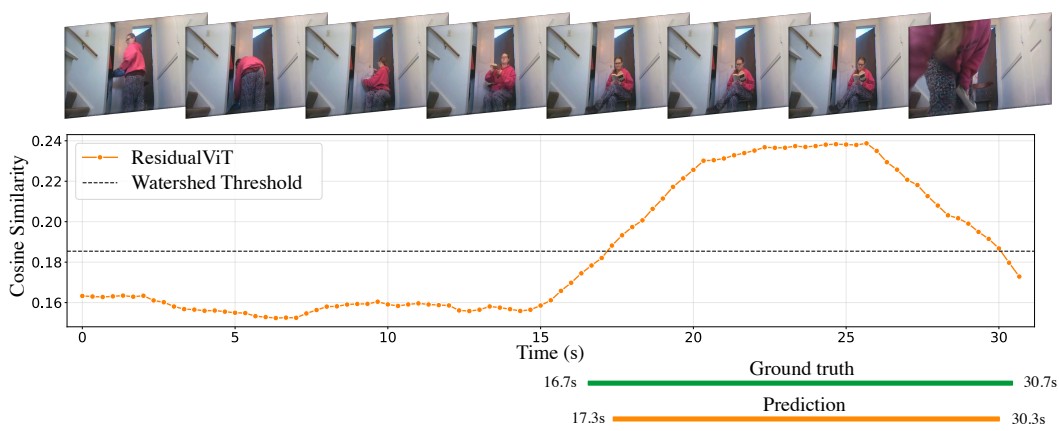

(b) **Grounding example.** We observe an IoU $= 0.93$ between the ground truth moment and the predicted one.

Figure 15: **Qualitative results.** We present two different examples in which our zero-shot algorithm can effectively ground the sentence in the video. We showcase the comparison between the ground truth annotation (**green**) and our top-1 prediction (**orange**).

## APPENDIX J  QUALITATIVE RESULTS OF NATURAL LANGUAGE VIDEO GROUNDING

In Figure 15-17, we present a series of qualitative results from the Charades-STA dataset, demonstrating the efficacy of our zero-shot grounding baseline in identifying relevant event boundaries within video content. In each example, we first show a subset of the video frames along with the textual query on top. Then, we illustrate the temporal sequence of similarity scores $\{S_t\}_{t=1}^{n_v}$ produced by computing the cosine similarity between each frame feature and the sentence feature. We also show the watershed threshold, which is used to determine the start and end moment predictions as detailed in Appendix C. For each example, the figure also illustrates the top-1 predicted temporal segment (**orange**) and the ground truth annotation (**green**).

In the examples depicted in Figures 15(a-b) and 16(a-b), our algorithm is capable of discriminating subtle frame differences and produces very precise temporal boundaries that provide an IoU $> 0.9$ with the ground truth. In example 15a, the feature representations of the frames and the sentence provide higher similarity when the television is present, in accordance with the query *"The person*

Query: The person is eating a sandwich.

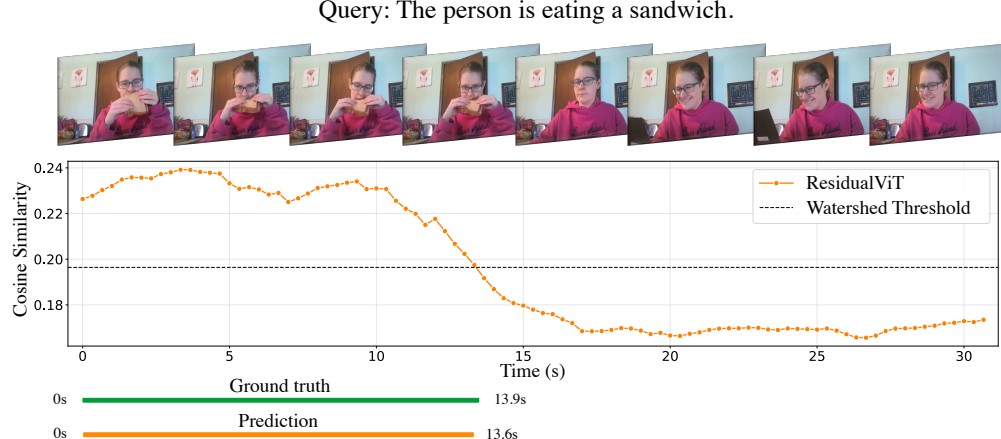

(a) **Grounding example.** We observe an IoU = 0.98 between the ground truth moment and the predicted one.

Query: A person is fixing a light.

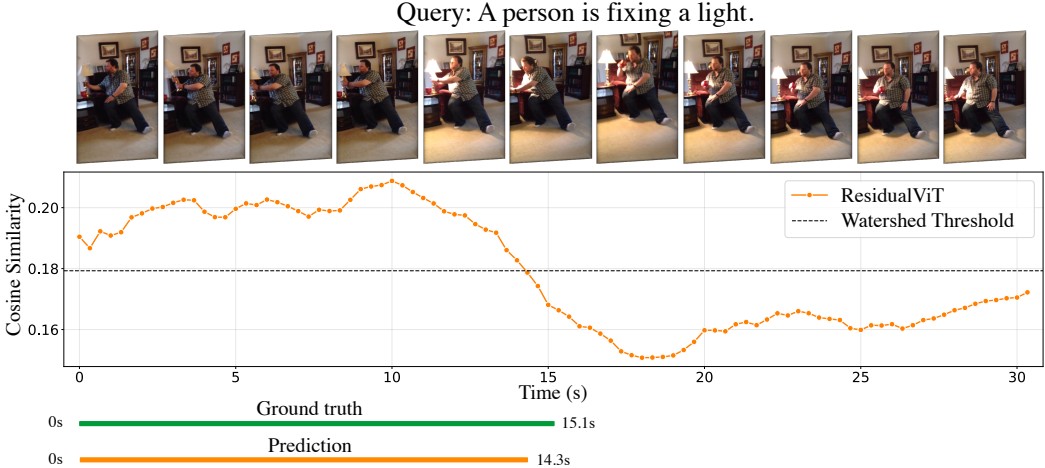

(b) **Grounding example.** We observe an IoU = 0.95 between the ground truth moment and the predicted one.

Figure 16: **Qualitative results.** We present two different examples in which our zero-shot algorithm can effectively ground the sentence in the video. We showcase the comparison between the ground truth annotation (**green**) and our top-1 prediction (**orange**).

*is watching television"*. Similarly, in example 15b, the algorithm can distinguish whether the person is holding a book despite the high resemblance among all frames, correctly predicting the temporal span relative to the textual query *"A person reads a book"* with IoU = 0.93. The cosine similarity profile in example 16a clearly differentiates between the section of the video in which the person is eating a sandwich and when they are simply smiling at the camera, predicting the grounding of the action *"A person is eating a sandwich"*, achieving IoU = 0.98. Example 16b presents a challenging scenario, *"A person is fixing a light"*, where the model needs to recognize the light's transition from off to on. Despite these complexities, our method provides a correct prediction with an IoU = 0.95.

Nonetheless, our approach can provide meaningful predictions that, however, do not align well with the ground truth moment. We detail one such example in Figure 17a. For the query *"A person puts a coffee cup on a shelf"*, we predict a temporal span that is correctly centered to the ground truth span but is twice as long as the ground truth moment, yielding an IoU of approximately 0.5. However, if we pay attention to the video frames, one could argue the prediction is still correct, as it begins when the person opens the cabinet and finishes after the person has placed the coffee cup in it.

Query: A person puts a coffee cup on a shelf.

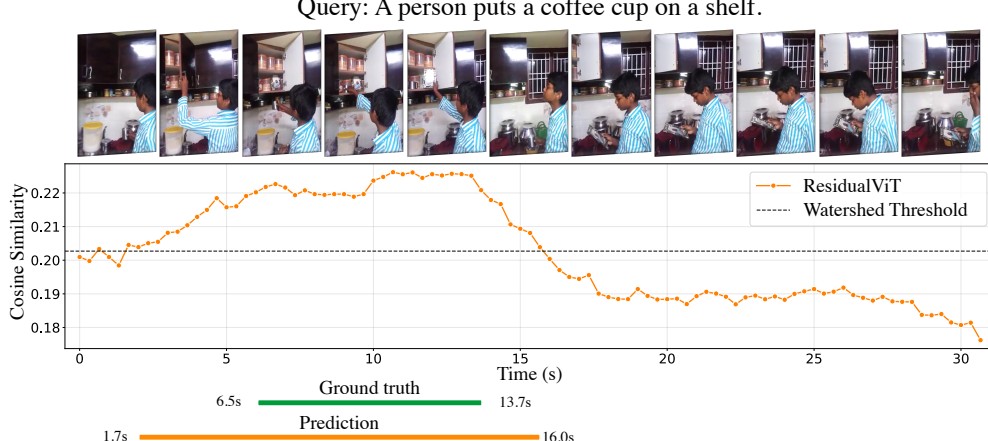

(a) **Grounding example.** An IoU = 0.5 is observed between the temporal annotation and our prediction.

Query: Person undressing by the shelf beside the doorway.

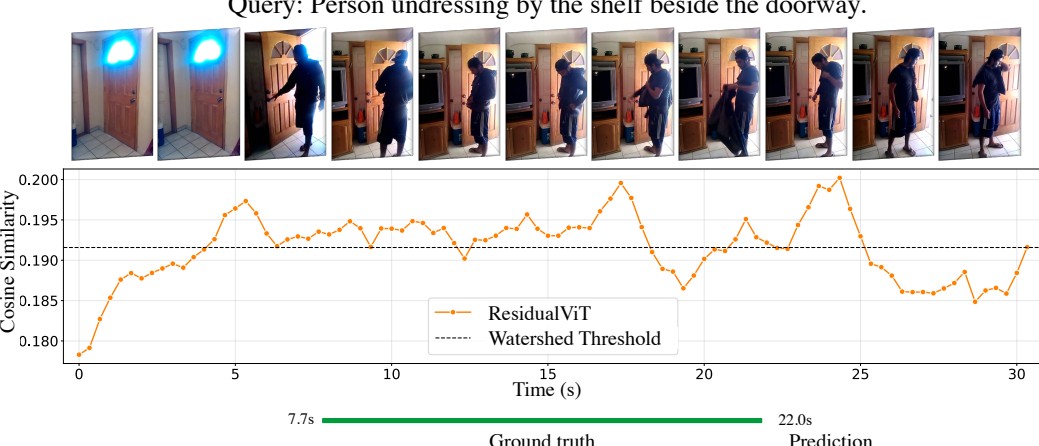

(b) **Grounding example.** Our prediction does not overlap with the ground truth moment.

Figure 17: **Qualitative results.** We present two different examples in which our zero-shot algorithm can effectively ground the sentence in the video. We showcase the comparison between the ground truth annotation (**green**) and our top-1 prediction (**orange**).

Lastly, in Figure 17b, we depict an example in which our proposed solution fails. The action described by the query *"Person undressing by the shelf beside the doorway"* shows a long duration, effectively producing a high similarity response for a good part of the video. This, in turn, affects the watershed threshold, which is proportional to the average similarity scores. Due to the high value of the threshold, our algorithm produces an incorrect prediction that does not overlap with the ground truth.

## APPENDIX K  FEATURE COMPARISON UNDER FULL SUPERVISION SETUP

In this section, we focus on a representative fully supervised baseline for Natural Language Temporal Video Grounding to evaluate the accuracy gap between CLIP and ResidualViT features. For this experiment, we selected CG-DETR (Moon et al., 2023), a recent and well-performing publicly available baseline that natively utilizes CLIP features for the Charades-STA dataset. The results of our experiments are presented in Table 7, and we maintained all hyperparameters as defined by the official implementation. Notably, features were extracted at a rate of one frame per second. For

| | Features | | R@1 | | mIoU | Avg. Cost Feature/sec (GFLOPs) |
|---|---|---|---|---|---|---|
| | | IoU=0.3 | IoU=0.5 | IoU=0.7 | | |
| CG-DETR | CLIP (B/32) | 63.6 | 49.7 | 26.8 | 43.8 | 4.4 |
| CG-DETR | ResidualViT (B/32) | 62.2 | 48.2 | 26.4 | 42.5 | $2.0_{(-53\%)}$ |
| CG-DETR | CLIP (B/32) + SlowFast | 69.6 | 57.1 | 34.5 | 49.0 | 40.5 |
| CG-DETR | ResidualViT (B/32) + SlowFast | 69.2 | 56.5 | 34.0 | 48.7 | 38.1 |
| CG-DETR* | CLIP (B/32) + SlowFast | 70.4 | 58.4 | 36.3 | 50.1 | 40.5 |

Table 7: **Frame Feature Comparisons in Full Supervision Setup.** This table compares the performance of the baseline CG-DETR (Moon et al., 2023) on the Charades-STA dataset under two setups: (i) using either CLIP (B/32) or ResidualViT (B/32) alone, and (ii) combining SlowFast features with either CLIP (B/32) (as in the original manuscript (Moon et al., 2023)) or ResidualViT (B/32). Our ResidualViT achieves a 53% reduction in frame encoding cost while closely maintaining the accuracy of the original setup. We denote with the symbol ∗ the accuracy as presented in the original paper (last row). Note that all other rows have been trained from scratch using the original codebase.

all rows except the last one, we train CG-DETR from scratch. The last row reports the accuracy as presented in the original paper. We find that we cannot fully reproduce those results using the default settings.

We begin by comparing the performance when using only CLIP features versus ResidualViT features, as shown in the first two rows of the table. For ResidualViT, we set $N=2$ and $p=85\%$. ResidualViT achieves a reduction in encoding cost of approximately 53% while maintaining accuracy close to the CLIP features. Specifically, we observe a marginal drop of 1.4% (relative 2.2%) for R@1-IoU=0.3, an absolute drop of 1.5% (relative 3.0%) for R@1-IoU=0.5, and an absolute drop of 0.4% (relative 1.5%) for R@1-IoU=0.7. These results indicate that, with an average relative accuracy drop of only 2.2%, we can achieve more than a 50% reduction in encoding cost.

Additionally, we evaluated the performance of CG-DETR in its original configuration, where CLIP features are channel-wise combined with SlowFast (Feichtenhofer et al., 2019) features. This setup significantly increases computational cost, as SlowFast features alone are estimated at 36.1 GFLOPs per feature. While the addition of SlowFast features can boost average accuracy on average of approximately 7.0%, it comes with a 9.2× increase in computational cost, representing an unfavorable trade-off. Nonetheless, when SlowFast features are combined with ResidualViT features, the computational cost is reduced by approximately 6%, with only a 0.5% absolute drop (relative 1%) in average accuracy, providing once again a favorable balance between accuracy and cost reduction.

## APPENDIX L   ADDITIONAL TASK: ACTION RECOGNITION

In this section, we evaluate the task of action recognition by examining the accuracy gap between CLIP and ResidualViT ($N = 2$, $p = 85\%$) features, using the ViT-B/32 backbone for both models. The experiments are conducted on the Kinetics-400 dataset (Kay et al., 2017) in a zero-shot setting.

The accuracy comparisons are presented in Table 8. In particular, we investigate the accuracy trends and total encoding costs as the number of frames increases.

We observe that ResidualViT delivers competitive accuracy compared to CLIP features, with a minimum gap of 0.8% for Accuracy@1 at 3 frames and a maximum gap of approximately 3.2% for Accuracy@1 at 4 frames. Similar trends are observed for Accuracy@5. However, when analyzing the accuracy versus total encoding cost, ResidualViT demonstrates a clear advantage: with 4 frames and a total cost of 12.8 GFLOPs, it outperforms CLIP with 3 frames and a total cost of 13.2 GFLOPs for both Accuracy@1 and Accuracy@5. Furthermore, ResidualViT with 7 frames and a total encoding cost of 21.2 GFLOPs achieves nearly identical accuracy to CLIP with 5 frames, which has a higher total cost of 22.0 GFLOPs.

| Number of Frames | CLIP | | Total encoding cost (GFLOPS) | ResidualViT | | Total encoding cost (GFLOPS) |
|---|---|---|---|---|---|---|
| | Acc@1 | Acc@5 | | Acc@1 | Acc@5 | |
| 1 | 44.5 | 72.3 | 4.4 | 44.5 | 72.3 | 4.4 |
| 2 | 45.0 | 73.0 | 8.8 | 43.4 | 71.1 | 6.4 |
| 3 | 43.9 | 71.5 | 13.2 | 43.1 | 70.7 | 8.4 |
| 4 | 48.1 | 76.0 | 17.6 | 44.8 | 73.0 | 12.8 |
| 5 | 46.5 | 74.5 | 22.0 | 45.1 | 73.5 | 14.8 |
| 6 | 48.7 | 76.8 | 26.4 | 45.5 | 73.9 | 16.8 |
| 7 | 47.6 | 75.9 | 30.8 | 44.4 | 72.9 | 21.2 |
| 8 | 49.3 | 77.1 | 35.2 | 46.5 | 74.9 | 23.2 |
| 9 | 48.2 | 76.7 | 39.6 | 46.2 | 74.8 | 25.2 |
| 10 | 49.3 | 77.4 | 44.0 | 46.5 | 75.1 | 29.6 |

Table 8: **Action Recognition.** We report accuracy at 1 (Acc@1) and accuracy at 5 (Acc@5) for CLIP and ResidualViT ($N = 2$, $p = 85\%$) features on the Kinetics 400 (Kay et al., 2017) dataset under a zero-shot setting.

This experiment demonstrates that our ResidualViT features are transferable to tasks beyond Natural Language Temporal Video Grounding (the main focus of our work) and Automatic Audio Description, as discussed in Section 4.2.

For the zero-shot setup of this experiment, each frame is encoded using either CLIP or ResidualViT, and the resulting visual feature representations are averaged. Classification is performed by combining the class labels with prompt templates provided by the CLIP baseline[1] and encoding the text using the language encoder. All prompt features per class are then averaged, and cosine similarity between the visual and text representations for each class is computed. The classes are ranked by their similarity scores, and the accuracy metric is computed accordingly.

## APPENDIX M    ADDITIONAL TASK: TEMPORAL ACTIVITY LOCALIZATION

In this section, we evaluate the task of temporal activity localization (TAL) by comparing the accuracy of CLIP and ResidualViT ($N = 2$, $p = 85\%$) features. Both models utilize the ViT-B/32 backbone, and the experiments are conducted on the ActivityNet dataset (Caba Heilbron et al., 2015).

For this experiment, we selected ActionFormer (Zhang et al., 2022) as a recent, high-performing, and easy-to-use baseline for TAL. The baseline is trained from scratch separately for each set of features, with features extracted at a rate of one feature per second. Our results show that CLIP features achieve a mean Average Precision (mAP) of 34.05%, while ResidualViT closely follows with an mAP of 33.25%. Importantly, this result demonstrates that ResidualViT, which operates at only 44% of the computational cost of CLIP, can deliver accuracy that is highly competitive with the CLIP upper bound on vision-only tasks.

---

[1]Prompt templates can be found here: https://github.com/openai/CLIP/blob/main/data/prompts.md#kinetics700

