# OpenReview forum: "ResidualViT for Efficient Zero-Shot Natural Language Temporal Video Grounding"
_ICLR.cc/2025/Conference — Submitted to ICLR 2025_

### Official Review · Reviewer_jGpQ · 2024-11-02

**Soundness:** 3
**Presentation:** 3
**Contribution:** 2
**Rating:** 6
**Confidence:** 4

**Summary:**

The paper proposes ResidualViT architecture for efficient temporal grounding of videos applicable to foundation vision-language models such as CLIP. Experiments are conducted on Charades-STA and ActivityNet-Captions dataset to show the effectiveness of the proposed method.

**Strengths:**

The paper proposes to reduce the cost of encoding video features by reducing the number of tokens and using residual connections. It shows reduction in GFLOPs by 50% on average across models. Ablation studies on the choice of each module shows the contribution of the proposed method.

**Weaknesses:**

Experiments are only conducted on only 2 datasets for the task of NLVTG. Experiments on MSR-VTT, TaCoS are necessary to analyze its generalization ability.

Missing evaluation results for Recall@5 although it is mentioned in the evaluation metrics section L340.


For token dropping, what about other pruning methods like DiffPruning [1] or Taylor Pruning [2] or Magnitude Pruning [3]?

[1] Fang et. al., Structural Pruning for Diffusion Models, NeurIPS 23

[2] Molchanov et. al. Pruning convolutional neural networks for resource efficient inference, ICLR 17

[3] He et. al., Channel pruning for accelerating very deep neural networks. ICCV 17


For ablation on the spatial resolution, the comparisons are made against ResidualVit token dropping strategy that is finetuned and so is not a fair comparison. What is the performance difference when finetuning adapters for those lower spatial resolution inputs?

**Questions:**

How about the performance when applying the method to supervised or weakly supervised methods?

For the runtime analysis, is the motion-based token reduction strategy applied? Computing the motion vectors adds to the latency and so the total time should include this for fair comparisons.

Why is the interleaving factor N set to 3 during training and is different for testing?

---

> ### Author Response · Authors · 2024-11-21
>
> **[W1] Additional tasks and datasets.** Thank you for raising this issue. We would like to clarify the choice of datasets and tasks. We purposely do not evaluate TaCoS and MSR-VTT for the following reasons. As also described in the answer to Reviewer y3Uz ([Q3]), we develop ResidualViT purposely for tasks requiring dense frame extraction, for which features are extracted in nearby frames with a certain degree of visual redundancy. Text-to-video retrieval is usually addressed by sparsely sampling 4-6 frames in videos, trying to reduce the redundancy among frames to obtain a more discriminative video representation [D]. This setting does not reflect the relevant use case for ResidualViT.
>
> Additionally, while TACoS has historically been popular, it no longer adequately captures the complexities needed to evaluate video grounding solutions. The TACoS dataset consists of only 27 videos for testing, totaling about 2.2 hours, all recorded with a static camera with an identical background, which limits its diversity. Additionally, the dataset focuses solely on cooking activities. The UniVTG [E] manuscript also notes that evaluating zero-shot approaches on TACoS is not very informative (Section 5.2.2).
>
> Instead, we have chosen to evaluate our model on **three** well-established datasets: ActivityNet Captions (4885 videos, total of 162 hours), Charades-STA (1334 videos, total of 12 hours), and MAD (112 videos, total of 210 hours). In particular, the challenging, long-form, large-scale, and highly diverse MAD dataset provides a robust benchmark for assessing our approach. By focusing on these datasets, we aim to provide a comprehensive evaluation that better reflects the diversity and dynamic nature of real-world scenarios, offering a more meaningful assessment of our model's capabilities.
>
> Additionally, with this rebuttal, we present results in two new tasks, namely: Action Recognition and Temporal Activity Localization. The results for Action Recognition are detailed in Appendix L (pages 30-31 of the updated manuscript), while results for Temporal Activity Localization are detailed in Appendix M (page 31 of the updated manuscript). For both tasks, we show that ResidualViT features provide an excellent accuracy vs. cost tradeoff, closely following the upper bound accuracy provided by CLIP but at a lower computational cost. Moreover, these results also show that our features can be successfully used for vision-only tasks.
>
> [D] Luo, Huaishao, et al. "Clip4clip: An empirical study of clip for end to end video clip retrieval." arXiv preprint arXiv:2104.08860 (2021).
>
> [E] Univtg: Towards unified video-language temporal grounding. In ICCV, 2023.
>
> **[W2] Metric.** We thank the reviewer for finding this typo. We would like to clarify that none of the weakly supervised, pseudo-supervised, and zero-shot methods cited in our work, except for CPL and U-VMR, report R@5 metrics. In line with previous works, we chose to focus on reporting the top-1 predicted moment, as it is considered the most relevant and important metric.
>
> However, for completeness, we have included the R@5 performance metric for the MAD dataset in Table 4 and commit to providing the R@5 results for Charades-STA and ActivityNet-Captions in the final version of the manuscript.
>
> **[W3] Additional References.** We thank the reviewer for these additional citations. As a result of this comment, we have added these references to the related work in Section 2.
>
> We would like to point out that these methods deal with model weights pruning/reduction, while our design aims at avoiding any modification to the weights of existing large-scale pre-trained models while reducing their inference cost by manipulating the input data fed to them. Therefore, the methodology proposed in these papers is complementary to our setting. Additionally, [1] and [3] do not do not address video understanding tasks.

---

> > ### Author Response · Authors · 2024-11-21
> >
> > **[W4] Fair comparison in frame resolution ablation.** We would like to clarify that in all experiments where the token reduction module is modified, we re-train the residual tokenizer to ensure consistent performance evaluation. No other parameter of the model is updated. Therefore, there is no finetuning of the transformer encoder.  **To enhance the clarity and scope of our work, we have modified Appendix E to reflect the information provided in this response.**
> >
> > Please note that our approach is designed to leverage large-scale pre-trained visual encoders with minimal architectural modifications (Lines 196-200). The requirement to fine-tune any layer of the visual encoder is precisely why the frame reduction approach is not optimal for our objectives; fine-tuning, especially in the earlier layers of the encoder, could significantly impact the rest of the architecture and diverge from our intended design philosophy.  Furthermore, adapting CLIP weights while preserving its performance and generalization properties remains an open research challenge requiring careful consideration, which is beyond the scope of our current work.
> >
> > As articulated in Lines 1221-1249 of the manuscript, reducing tokens proves to be a more effective strategy for preserving performance while minimizing modifications to model weights and architecture.
> >
> > **[Q1] Fully supervised method.** We identify CG-DETR as a recent fully supervised baseline. We select such a baseline for its competitive performance, the availability of code, and the native use of CLIP B/32 features. The objective of this investigation is to replace the existing CLIP features with the ResidualViT ones and train the grounding model from scratch on the Charades-STA dataset, comparing the final results.
> >
> > Similarly to all previous experiments and ablations, we find that ResidualViT features obtain close accuracy to the original CLIP features at a lower computational encoding cost. The results indicate that, with over 50% reduction in encoding cost, the average relative performance drop is only 2.2%. We report a complete description of our findings in a newly written Appendix K (page 30 of the updated manuscript).
> >
> > Note that in this section, we do not explore hyperparameters and only use the default ones, as provided by the official implementation, for all evaluated settings in Table 7 (Appendix). Features are extracted at a constant rate of one feature per second in accordance with the original codebase. The ResidualViT model used is B/32, N=2, p=85%.
> >
> > **[Q2] Runtime Analysis.** We would like to clarify that we are not computing optical flow but just reading the motion vectors present in the mp4 file. This step is performed concurrently with the sequential read of frames from the video and has a negligible time overhead. We refer the reviewer to Appendix B of the manuscript, where we provide the implementation details regarding the motion vector processing.
> >
> > Also, note that the runtime analysis only measures the forward pass time and does not account for dataloading. Nonetheless, the computational overhead of the video reader module [AcherStyx] is minimal (~0.04 GFLOP per frame against the 77.8 GLOPs required by the ViT-L/14 model), as it involves a sequential pass over the video data with minimal processing required to format the motion vectors, which are stored as is in the file and merely need to be organized into a matrix format.
> >
> > **[Q3] Interleave Factor During Training.** We sincerely thank the reviewer for this insightful suggestion. We are actively working on this specific experiment of training with varying N values and will do our utmost to complete it within the rebuttal period.

---

> > > ### Comment · Reviewer_jGpQ · 2024-11-25
> > >
> > > Thank you to the authors for the detailed rebuttal. It addressed most of my concerns. Below are my comments
> > >
> > > Regarding the frame resolution ablation question, I was referring to fine-tuning with LoRA adapters, as this approach does not modify the encoder, making it a more fair comparison.
> > >
> > > The authors state that existing pruning methods are complementary but do not provide experimental validation to support this claim. Given the paper's focus on efficiency, it should include comparisons with existing pruning methods which would provide a clearer assessment of the method's effectiveness.
> > >
> > > Additionally, for the fully supervised baseline, the reported results differ from those presented in the CG-DETR paper (CLIP+Slowfast baseline). Could you clarify the reason for this discrepancy?

---

> > > > ### Author Response · Authors · 2024-11-25
> > > >
> > > > We thank the reviewer for the timely response. Here are our clarifications regarding the remaining concerns.
> > > >
> > > >
> > > >
> > > > **[Q3] Interleave Factor During Training:** We sincerely thank the reviewer for this insightful suggestion. In response, we have conducted a new ablation study on the N_Train hyperparameter, and the results are included in Appendix E (pages 25-26) of the updated manuscript.
> > > >
> > > > In this ablation, we trained ResidualViT using different values of N_Train (specifically N_Train = 3, 5, and 10). Moreover, we investigate two distinct frame sampling strategies at training time. Each trained model was then tested on the Charades-STA dataset with varying inference interleave factors (N=1,2,3,5,10). For this experiment, we used the ViT-B/32 backbone, which differs from the Figure 4 ablation that evaluates accuracy vs. cost using the ViT-L/14 backbone. This choice was made to ensure timely completion within the rebuttal deadline, as the ViT-B/32 backbone is significantly faster to train (4.4 GFLOPs per frame compared to 77.8 GFLOPs for ViT-L/14). Our observations indicate that all training configurations produce similar accuracy trends, this means that increasing the number of frames in training or modifying the sampling strategy does not bring accuracy improvements on the downstream zero-shot evaluation.  Based on our previous results, we hypothesize that the observed trends will generalize across backbones and are happy to extend this analysis to ViT-L/14 for the camera-ready version of the manuscript.
> > > >
> > > > **[Q4] LoRa.** We sincerely thank the reviewer for suggesting this experiment. While we fully acknowledge the benefits of LoRa adapters for minimally invasive fine-tuning. We are actively working on this specific experiment where we use LoRa adapters for finetuning the 2D convolutional operation that implements the frame pacthyfication and will do our utmost to complete it within the rebuttal period.
> > > >
> > > > **[Q5] Scope.** We sincerely thank the reviewer for the suggestion. We would like to point out that our approach explicitly avoids modifying the visual encoder, as detailed in Lines 196–200 and elaborated in Lines 1287–1293 of the manuscript. This constraint is central to our approach as modifications to existing large-scale pre-trained network weights require careful investigation. Therefore, our approach leverages pre-trained models as-is, utilizing the ViT design to reduce computational costs for video encoding. This design is motivated by the moderate redundancy in consecutive video frames, which leads to computational inefficiencies in video encoding. Our method reduces costs by modifying the input to the network rather than altering its weights.
> > > >
> > > > In the broader field of deep learning, non-specialized approaches for cost reduction—such as (i) distillation, (ii) pruning, and (iii) quantization—focus on reducing the computational demands of a network by manipulating its weights. These methods are fundamentally different from ours, as (1) our approach is video-specific, (2) it targets temporally dense tasks, (and 3) it operates solely on the input data to the visual encoder.
> > > >
> > > >
> > > > These methods are not competitors to ours but rather complementary techniques. They can be applied in addition to our framework, offering further cost savings on top of the residual tokenizer and token reduction module. However, exploring such combinations falls outside the current scope and constitutes a promising direction for future work, as it does not directly affect our claims or results.
> > > >
> > > > Furthermore, we note that the papers mentioned by the reviewer address convolutional networks or unrelated tasks (i.e., image generation, image classification, object detection), making their direct application to our framework non-trivial. Exploring their relevance to our video-specific setting and transformer-based architecture would require significant adaptation and evaluation, which is beyond the focus of this work.
> > > >
> > > > We hope this explanation clarifies the distinctions between these methods and our approach.

---

> > > > > ### Author Response · Authors · 2024-11-25
> > > > >
> > > > > **[Q6] Performance mismatch.** We thank the reviewer for highlighting this discrepancy. To ensure a fair and accurate comparison, we trained CG-DETR from scratch for each row presented in Table 7 using the official codebase provided by the authors and the default hyperparameters. The difference in results for line 3 of Table 7 compared to those reported in the CG-DETR paper suggests that the original results may not be fully reproducible under the default settings.
> > > > >
> > > > > We acknowledge that tuning hyperparameters could potentially improve accuracy across all settings. However, due to time constraints during the rebuttal period, this was not feasible. Importantly, this discrepancy does not affect the main conclusion of the ablation study: **our ResidualViT features maintain highly competitive accuracy with respect to the CLIP upper bound while significantly reducing computational costs.**
> > > > >
> > > > > For completeness, we added the original performance as an additional row and discussed the reproducibility issue in the text. We appreciate the opportunity to clarify this point and hope this explanation resolves any concerns.

---

> > > > > > ### Author Response · Authors · 2024-11-26
> > > > > >
> > > > > > **[Q4] LoRa.** We are pleased to inform the reviewer that we have conducted the requested experiment and updated Figure 10 and the corresponding text (Lines 1285–1291) with the new results.
> > > > > >
> > > > > > Our findings indicate that LoRa adapters consistently improve the performance of ResidualViT with lower-resolution inputs across all interleave factor (N) values, with accuracy gains ranging from 0.5% to 2.6% for R@1-IoU=0.5. These results suggest that ResidualViT with reduced input resolution achieves a competitive accuracy vs. cost tradeoff compared to ResidualViT equipped with token dropping.
> > > > > >
> > > > > > However, token dropping continues to offer better accuracy while maintaining the distinct advantage of requiring no updates to network weights.

---

### Official Review · Reviewer_HVcE · 2024-11-04

**Soundness:** 3
**Presentation:** 3
**Contribution:** 2
**Rating:** 6
**Confidence:** 5

**Summary:**

This paper proposes a novel method for zero-shot temporal grounding, which differs from traditional approaches that typically focus on reducing model parameters. The authors utilize the inherent redundancy between frames to transfer the ViT model from the image domain to the video domain. Additionally, they design a distillation mechanism for parameter learning. The approach is validated on three datasets, demonstrating significant reductions in computational costs compared to the baseline.

**Strengths:**

- The authors efficiently compute video features by leveraging the inherent temporal redundancy in videos. While this is a common practice in the video domain, its application to zero-shot temporal video grounding is relatively novel.
- After pre-training, the proposed method does not require training on the target dataset. This allows for direct application across different video grounding datasets, providing good flexibility.

**Weaknesses:**

- There are concerns regarding the comparison with other zero-shot methods. As a reviewer for this paper's NeurIPS 2024 submission, I noted significant differences in GFLOPs between versions. Specifically, for the Charades-STA and ActivityNet-Caption datasets, the GFLOPs were reported as 148.4 and 9.6 for the NeurIPS version, while the current submission shows values of 1370 and 370, respectively. The authors should clarify the testing process and conditions for GFLOPs to help us understand the substantial discrepancies. Since the paper emphasizes efficiency, it is crucial to compare both performance and computational metrics accurately. Additionally, could the authors provide comparisons of throughput across different methods?
- Tied to the above question, is the frame sampling rate consistent when comparing the computational costs of different methods?

**Questions:**

- Please clarify the specific computation process and methodology used for calculating the computational costs.
- Although the proposed method does not require training on the target video grounding dataset, it is important to note that the distillation process utilizes the WebVid-2.5M dataset, which effectively uses more video-text pairs than other methods. Is this comparison sufficiently fair?
- Since the proposed method requires pre-training on a video-text dataset, would using more pre-training data yield better results? What is the scalability of the method?
- If the parameter 𝑁 is increased from 3 to a larger value, how would the performance be affected?

---

> ### Author Response · Authors · 2024-11-21
>
> **[Q1] Computational costs clarification.** We thank the reviewer for the opportunity to discuss this key aspect. The discrepancy between the previous and current versions of our computational cost analysis stems from our misinterpretation of the prior work, MR-FLVM. Prior to our paper resubmission to ICLR’25, we conducted a careful re-analysis of MR-FLVM and engaged in direct discussions with the MR-FLVM authors via email.
>
> MR-FLVM represents video data as 32 snippets, each containing a few frames, which are initially encoded using 3D networks (C3D at 38.5 GFLOPs for Charades-STA and I3D at 148.4 GFLOPs for ActivityNet-Captions). In addition, each snippet is re-encoded using InternVideo for snippet-text similarity calculations—contrary to our initial assumption that this step used CLIP. The use of InternVideo, which applies full self-attention over all patches in 16 frames using a large model (InternVideo-MM-L-14), incurs a significant computational cost of 1.332 TeraFLOPs per feature (measured by us on the official InternVideo implementation). Once similarity scores are obtained, each snippet feature is refined through a weighted sum of all snippet representations with high similarity scores to the text query, effectively smoothing the snippet representations over time. The snippets are then combined into proposals and re-scored using InternVideo.
>
> Based on this information, we estimated the average feature extraction rate for Charades-STA to be ~1 feature per second (as the average video duration is 30 seconds) and ~0.25 features per second for ActivityNet-Captions (with average video durations of 2 minutes).
>
> The total average computational cost per feature per second, which we use as a metric in Table 2, is calculated as the cost of encoding one feature multiplied by the extraction rate of features per second. For MR-FLVM, this results in a computational cost of (38.5 + 1332) GFLOPs × 1 = 1370 GFLOPs for Charades-STA, and (148.4 + 1332) GFLOPs × 0.25 = 370 GFLOPs for ActivityNet-Captions.
>
>
> **[Q2] Fair comparison.** We appreciate the opportunity to address this concern. The primary purpose of the distillation loss (Equation (2)) is to train the residual tokenizer to approximate the original CLIP features. This process focuses on feature alignment rather than task-specific optimization, i.e., our training loss does not explicitly encourage the feature representation to learn new information beyond the original CLIP representation. Note that the accuracy of our approach is upper-bounded by the original CLIP representation.
>
> Further evidence is provided in the ablation regarding the distillation strategy presented in Appendix E. In detail, we remove the language features from the distillation approach and replace the loss to be the mean-squared error (MSE) between the visual representation of the ViT teacher and the visual representation of the ResidualViT student. We illustrate the new distillation pipeline in Figure 11(a). Our investigation results are reported in Figure 11(b) where we showcase that this approach provides near identical grounding accuracy as compared to the distillation that adopts language features and cross-entropy loss (Equation (2)). Notice that for a fair comparison, all training and inference hyperparameters are kept exactly the same. The key takeaway is that ResidualViT is capable of closely approximating the CLIP features and retaining the alignment with the original CLIP language model even in this setting.
>
>
> **[Q3] More pre-training data and scalability.**  We would like to clarify that our method involves only one stage of training of the residual tokenizer alone, i.e., there is no post-training stage. It is conceivable that incorporating additional data points could further enhance the quality of distillation. However, our accuracy already approaches the upper bound established by CLIP, which suggests that further scaling of the training data may yield diminishing returns. One of the key reasons is that the number of learnable parameters is small (~400K for B/32 and B/16 models and ~800K for L/14), as only the residual tokenizer weights are learned.
>
> **[Q4] Increasing N to a larger value.** We would like to clarify that the accuracy vs. cost trade-off for different values of the interleaving factor N is analyzed in Figure 4 of the main paper, where we report accuracy for N=1,2,3,5, and 10. Additionally, in response to requests from reviewers cDMc (Q1) and jGpQ (Q3), we are actively conducting experiments on training with varying N values and will make every effort to complete these experiments within the rebuttal period.

---

> > ### Comment · Reviewer_HVcE · 2024-11-25
> > **Request for further clarification**
> >
> > Thank you for your detailed explanations regarding the computational costs; they have significantly clarified the discrepancies in GFLOPs between versions and enhanced my understanding of your methodology.
> >
> > However, I noticed that you did not address my question about throughput comparisons across different methods. Since efficiency is a key focus of your paper, providing throughput comparisons would be valuable for understanding the practical implications of your approach. Could you please provide such comparisons or elaborate on why they might not be applicable?
> >
> > Regarding your statement that "the accuracy of our approach is upper-bounded by the original CLIP representation," I find this point a bit unclear. CLIP, as an image-text model, does not inherently handle temporal data. Your method appears to extend CLIP's capabilities by enhancing its temporal modeling through the residual tokenizer. Therefore, why is the performance of your approach considered upper-bounded by the original CLIP representation? Could you please clarify this aspect?
> >
> > Additionally, I feel that my question about the fairness of the comparison was not directly addressed. Specifically, since your distillation process utilizes the WebVid-2.5M dataset, which effectively employs more video-text pairs than other methods, is the comparison sufficiently fair? If you were to use the same amount of data as other methods, how would the performance of your method be affected? Understanding this would help assess the scalability and generalizability of your approach.

---

> > > ### Author Response · Authors · 2024-11-25
> > >
> > > **[Q4] Interleave Factor During Training.** We sincerely thank the reviewer for this insightful suggestion. In response, we have conducted a new ablation study on the N_Train hyperparameter, and the results are included in Appendix E (pages 25-26) of the updated manuscript.
> > >
> > > In this ablation, we trained ResidualViT using different values of N_Train (specifically N_Train = 3, 5, and 10). Moreover, we investigate two distinct frame sampling strategies at training time. Each trained model was then tested on the Charades-STA dataset with varying inference interleave factors (N=1,2,3,5,10). For this experiment, we used the ViT-B/32 backbone, which differs from the Figure 4 ablation that evaluates accuracy vs. cost using the ViT-L/14 backbone. This choice was made to ensure timely completion within the rebuttal deadline, as the ViT-B/32 backbone is significantly faster to train (4.4 GFLOPs per frame compared to 77.8 GFLOPs for ViT-L/14). Our observations indicate that all training configurations produce similar accuracy trends, this means that increasing the number of frames in training or modifying the sampling strategy does not bring accuracy improvements on the downstream zero-shot evaluation.  Based on our previous results, we hypothesize that the observed trends will generalize across backbones and are happy to extend this analysis to ViT-L/14 for the camera-ready version of the manuscript.
> > >
> > > **[Q5] Throughput.** We would like to request further clarification from the reviewer regarding the requested experiments.
> > > If the reviewer is interested in comparing the throughput of different backbones for feature extraction, we would like to highlight that we have already investigated the throughput gains of ResidualViT compared to a standard ViT in Appendix F: ResidualViT Runtime. As shown in Figure 14, ResidualViT achieves not only a significant reduction in GFLOPs but also a 2.5x improvement in encoding time. This translates to more than double the feature extraction throughput compared to a standard ViT.
> > >
> > > Alternatively, if the reviewer is requesting throughput comparisons for the pseudo-supervised methods presented in Table 2, we would like to clarify that five of these methods (U-VMR, PZVMR, CORONET, LFVL, MR-FVLM) do not provide publicly available code. Of the three that do (PSVL, SPL, UniVTG), only SPL provides competitive accuracy, making it the most suitable candidate for comparison. If the reviewer deems it relevant, we would be happy to measure SPL’s throughput in terms of grounded queries per second on the Charades-STA dataset.
> > >
> > > Additionally, we would like to emphasize that our zero-shot grounding algorithm is fundamentally more efficient, as it only requires cosine similarity computations over frames and language representations without the need for forward passes through a grounding network. For instance, our inference requires just ~10.8 seconds to process the grounding of 3720 queries across 1334 videos, averaging approximately 3 ms per query in the Charades-STA dataset.
> > >
> > > We kindly ask the reviewer to advise us on the most relevant experiment or comparison they would like to see.
> > >
> > > **[Q6] Clarification of the upper bound terminology.** We are happy to clarify why we use the upper-bound terminology for CLIP in our approach.
> > >
> > > As illustrated in Figure 2 of the main paper, our approach uses an interleaved frame encoding strategy. Specifically, for a sparse set of I-feature locations, I-features are computed directly using the original CLIP ViT encoder. While, for a dense set of P-feature locations, features are generated using the ResidualViT encoder.  ResidualViT is composed of three modules, (i) a ViT encoder initialized with the same weights as CLIP, (ii) a token reduction module to reduce the number of input tokens to the encoder and, therefore reduce the encoding cost (iii) a residual tokenizer which transforms the nearby I-features to the input space of the ViT encoder. This module is used as nearby frames have visually similar information that can be used to recover the missing information induced by the token reduction mechanism.
> > >
> > > **The objective of ResidualViT is to approximate the original CLIP features at the P-feature locations at a lower computational cost.** Ideally, this means perfectly reconstructing the CLIP features for those frames. To achieve this, we designed a distillation training (Section 3.2) used to learn the parameters of the residual tokenizer. To achieve this, we leverage video data to conditionally reconstruct the P-feature representations based on the nearby I-feature locations.
> > >
> > > **In summary, since our method is trained to approximate CLIP representations, we use the terminology that ResidualViT is upper-bounded by CLIP.** We are happy to use a different terminology if the reviewers deem it more appropriate.

---

> > > > ### Author Response · Authors · 2024-11-25
> > > >
> > > > **[Q7] Fairness.** Thank you for offering us the opportunity to clarify this important point. For the pseudo-supervised methods presented in Table 2, we would like to note that they are training one specific set of network weights for each target dataset. In contrast, we train one single network (per backbone size) and address four different tasks with the same model weights. To clarify the pseudo-supervised methods cannot transfer from one dataset to another for the same task.
> > > >
> > > > Moreover, we would also like to point out that our distillation training can be carried out with videos only, without the need for paired text data, as outlined in Appendix E, with very similar accuracy. Therefore, effectively our approach does not require any paired video-text data and can be trained using only videos without the need for any manual language annotations.
> > > >
> > > > **Please note that our training aims at learning (distilling) the residual tokenizer such that ResidualViT can effectively approximate CLIP (as discussed in Q6) while the pseudo-supervised methods are training for the downstream task. Therefore directly comparing data scales is not straightforward.** We report the sizes of the different downstream datasets in lines 354-364 of the main paper. We will also clarify there the size and the type of annotations for the WebVid2.5M dataset used for distillation. We would be happy to incorporate any other suggestions.

---

> > > > > ### Comment · Reviewer_HVcE · 2024-11-29
> > > > >
> > > > > Based on the author's responses and feedback from other reviewers, I believe the paper does demonstrate novelty. Although the comparison with existing methods isn't entirely fair in terms of data scale (as mentioned in Q7), the clarification provided there addresses my concerns. The proposed method shows promise and has potential for further exploration, particularly with respect to its efficiency and scalability.

---

> > > > > > ### Author Response · Authors · 2024-11-29
> > > > > >
> > > > > > We sincerely thank the reviewer for their thorough review, as well as for their constructive feedback, which helped us improve our manuscript. We greatly appreciate the recognition of the novelty and potential of ResidualViT.

---

### Official Review · Reviewer_cDMc · 2024-11-06

**Soundness:** 3
**Presentation:** 3
**Contribution:** 3
**Rating:** 8
**Confidence:** 4

**Summary:**

This paper presents an efficient method for feature extraction from dense video frames. The key idea is to exploit temporal redundancy in video. Features are extracted using vanilla ViT intermittently from keyframes. These features are subsequently propagated to nearby frames, whose features are extracted using a modified ViT, dubbed ResidualViT, with significantly reduced computational overhead. This is achieved by subsampling the input token embeddings while also inputting the features from the preceding keyframe to fill in the missing information. ResidualViT is trained using a distillation objective, with the goal of matching both the visual and textual features from a pre-trained CLIP. The experiments demonstrate that the method achieves competitive accuracy on the downstream task of video grounding yet higher efficiency when compared to running vanilla ViT on all frames.

**Strengths:**

- Strong motivation. The paper targets a key challenge in video understanding, namely the high computational cost of feature extraction from dense video frames for dense labeling tasks.
- Simplicity. The method leverages pre-trained models and its training does not require large data or compute. It thus is more practical and can be more easily adapted for a wide array of video understanding tasks.
- Effectiveness. Despite its simplicity, the method demonstrates strong zero-shot video grounding results.
- Clear writing. The paper is easy to follow and clearly describes the motivation, key intuition, and key components of the method.

**Weaknesses:**

- The method is highly dependent on model architecture. Token dropping and residual feature passing assume the ViT architecture, which operates on single images. It thus does not allow the modeling of temporal dependency. Further, the distillation loss requires features from both image and text encoders (CLIP). Finally, this somewhat restrictive setting is derived from the video grounding task, and does not seem to generalize on other video understanding tasks, where a text branch is not necessary. I am wondering whether applying a similar approach to ViT (image branch only) would work for tasks like temporal action localization that do not take text as input.

- Some key ablation studies are missing. Token reduction probability p is a free parameter that controls the computational cost of ResidualViT. It would be helpful to identify the optimal p that achieves the best efficiency-accuracy tradeoff. Also, it would be interesting to study whether matching text features in model distillation necessarily boosts video grounding accuracy. Finally, one may compare the N=1 (or N=2) setting with naive frame subsampling, that is, only extracting I-frame features (and skipping P-frames) for video grounding. There is no need to run ResidualViT in the latter case, resulting in higher efficiency than the proposed method.

**Questions:**

- According to Figure 4, the spacing N has to be kept small to avoid significant accuracy drop. I am wondering whether this can be explained by the training procedure, where training samples are features from neighboring frames. What if the spacing between the training pairs is randomly sampled from some interval?

---

> ### Author Response · Authors · 2024-11-21
>
> **[W1] Clarifications:** Thank you for your comments. We would like to address the points raised by the reviewer and clarify several aspects of our work:
> 1. We focus on transformer-based architectures as they have become the de-facto standard architecture in the vast majority of computer vision and natural language processing applications. The ViT architecture has proven to be highly scalable and well-performing. In this regard, we base our work on the most widespread backbone to date.
>
> 2. The reviewer's assessment is correct that ResidualViT does not explicitly model temporal relationships between frames. Nonetheless, our token reduction plus residual strategy is agnostic to the backbone type, allowing for flexibility in its application. For example, our approach could be extended to a spatiotemporal video foundation model by applying tokenization to 3D spatiotemporal volumes while still applying the token reduction module (i.e., dropping patches over volumes) and utilizing the residual tokenizer. **Our primary objective remains to develop a computationally efficient model** through distillation that approximates the accuracy of the foundation model (CLIP in our study). **Our research aims to provide an efficient framework for video frames encoding, complementing methods focused on temporal dependency modeling.** Moreover, as discussed in Section 2 (lines 104-115), frame-based video encoding remains popular due to its computational efficiency. While dedicated temporal modeling can enhance accuracy, it often increases computational demands, which must be carefully balanced against possibly modest accuracy gains.
>
> 3. We would like to draw the reviewer’s attention to Appendix E, where we present an ablation study of our distillation approach without requiring language. In detail, we remove the language features from the distillation approach and replace the loss to be the mean squared error (MSE) between the visual representation of the ViT teacher and the visual representation of the ResidualViT student. We showcase the new distillation pipeline in Figure 11(a). Our investigation results are reported in Figure 11(b), where we showcase that this approach provides near identical grounding accuracy as compared to the distillation that adopts language features and cross-entropy loss. Notice that for a fair comparison, all training and inference hyperparameters are kept exactly the same. **This result further demonstrates the flexibility of our methodology.**
>
> 4. Finally, as requested, we investigate the quality of our ResidualViT features as opposed to CLIP features on the task of Temporal Activity Localization. For this experiment, we identify ActionFormer [B] as a recent, well-performing, and easy-to-use baseline for the task. We extract features using the CLIP B/32 and ResidualViT B/32 (N=2, p=85%) visual backbones at one frame per second and train the ActionFormer baseline from scratch for both sets of features. We find that CLIP features can obtain a mAP of 34.05% while ResidualViT closely follows with a mAP of  33.25%. **This experiment confirms that our features, which can be computed with only 44% of the CLIP cost, can achieve very competitive accuracy with the CLIP upper bound on vision-only tasks.** We include this new result in Appendix M (page 31 of the updated manuscript).
>
> We also would like to point out that we performed a comparative study for the task of Action Recognition following reviewer y3Uz question 3, and we report the results in Appendix L (pages 30-31 of the updated manuscript). In this evaluation, we showcase how ResidualViT provides competitive performance with respect to CLIP for the same amount of encoded frames. Moreover, when analyzing the accuracy versus total encoding cost, ResidualViT demonstrates a clear advantage: with four frames and a total cost of 12.8 GFLOPs, it outperforms CLIP with three frames and a total cost of 13.2 GFLOPs for both Accuracy@1 and Accuracy@5.
>
> Moreover, following the request of reviewer cDMc, we employ the ResidualViT features in a fully supervised method for natural language temporal video grounding. We present the result in Appendix K, where we showcase that the ResidualViT features induce a cost saving of over 50% at the cost of an average relative accuracy degradation of about 2%.
>
> **All of these experiments point to the conclusion that our methodology has proven successful in reducing the video encoding cost while retaining close performance to the upper bound provided by CLIP for several video recognition tasks that now include Natural Language Temporal Video Grounding, Automatic Audio Descriptions, Action Recognition, and Temporal Activity Localization on four datasets, namely: Charades-STA, ActivityNet-Captions / ActivityNet, MAD, and Kinetics-400.**
>
> [B] Zhang, Chen-Lin, Jianxin Wu, and Yin Li. "Actionformer: Localizing moments of actions with transformers." European Conference on Computer Vision. Cham: Springer Nature Switzerland, 2022.

---

> > ### Author Response · Authors · 2024-11-21
> >
> > **[W2] Ablations.** Thank you for highlighting these ablations. Please find the answers below.
> >
> > - **Token reduction probability.** If we understand the reviewer correctly, we believe the requested ablation is already included in our work. We kindly refer the reviewer to Appendix E, where we present a detailed ablation study on the token reduction probability. Figure 8 illustrates the performance trend of CLIP versus ResidualViT as the token drop probability p is varied. Our findings indicate that ResidualViT maintains strong resilience to increasing token reduction probabilities up to p=85%. Beyond this point, performance begins to degrade. Based on these observations, we have selected p=85% as the default value in our approach. Please note that we explicitly reference this experiment in Section 4.1 while enumerating the various ablations provided in the appendix.
> >
> > - **Language during distillation.** While our distillation design includes both visual and language representations, we ablate the distillation strategy in Appendix E. In detail, we remove the language features from the distillation approach and replace the loss to be the mean-squared error (MSE) between the visual representation of the ViT teacher and the visual representation of the ResidualViT student. We illustrate the new distillation pipeline in Figure 11(a). Our investigation results are reported in Figure 11(b) where we showcase that this approach provides near identical grounding accuracy as compared to the distillation that adopts language features and cross-entropy loss (Equation (2)). Notice that for a fair comparison, all training and inference hyperparameters are kept exactly the same. The key takeaway is that ResidualViT is capable of closely approximating the CLIP features and retaining the alignment with the language model even in this setting.
> >
> > - **Frame subsampling ablation.** As requested, we conducted a frame subsampling ablation in which we reported performance using only the I-features. In particular, we evaluate ResidualViT (N=2 p=85%) at operating at 3 frames per second (one I-feature and two P-features) and CLIP at one frame per second on the Charades-STA dataset. ResidualViT achieves an accuracy of 34.2 at R@1-IoU=0.5 and   17.7 at R@1-IoU=0.7 for an average encoding cost of 6.1 GFLOPs, while CLIP accuracy is 30.1 at R@1-IoU=0.5 and 13.9 at R@1-IoU=0.7 for a cost of 4.4 GFLOPs.
> > Inspired by this comparison, we perform a more comprehensive analysis of the accuracy obtained by the two methods for varying encoding frames-per-second (FPS). This experiment highlights that ResidualViT provides superior accuracy-to-cost trade-offs, even when CLIP operates at a lower framerate. Specifically, at an equivalent computational budget of approximately 4 GFLOPs, ResidualViT achieves over 3% higher accuracy for R@1-IoU=0.5 compared to CLIP. **We include this new ablation in Appendix E (page 25 of the updated manuscript).**
> >
> > **[Q1] Interleave Factor During Training.** We sincerely thank the reviewer for this insightful suggestion. We are actively working on this specific experiment of training with varying N values and will do our utmost to complete it within the rebuttal period.

---

> ### Comment · Reviewer_cDMc · 2024-11-25
>
> I would like to thank the authors for their detailed response. I acknowledge that some of the requested ablations were previously included in the appendix and am glad to see the results further highlight the strength of the proposed method. For this reason, I have raised my rating and recommend acceptance of the paper. In the meantime, I encourage the authors to reconsider the title of the paper. As the other reviewers correctly pointed out, the proposed approach is not limited to video grounding and has been shown to be effective for other video understanding tasks. Please revise the text accordingly to not overemphasize ResidualViT as a video grounding method.

---

> > ### Author Response · Authors · 2024-11-29
> >
> > We sincerely thank the reviewer for their thorough review, which has greatly helped us improve our manuscript. We deeply appreciate the recommendation for acceptance. Regarding the title, we propose “ResidualViT for Efficient Video Encoding” as a suitable candidate and will ensure the camera-ready version reflects all feedback.

---

### Official Review · Reviewer_y3Uz · 2024-11-06

**Soundness:** 2
**Presentation:** 1
**Contribution:** 2
**Rating:** 3
**Confidence:** 5

**Summary:**

This paper presents a novel framework, named ResidualViT, to efficiently transfer a pretrained image model (i.e., ViT) to video data for zero-shot natural language temporal video grounding.
Given that the number of patches per frame directly impacts processing costs when applying an image transformer across multiple video frames, the framework addresses this by categorizing video frames into two types:
To address this challenge, the proposed method divides video frames into two types:

1) I-frames: Processed by the original image encoder to extract comprehensive I-features using all available tokens, capturing full information;
2) P-frames: Processed by the ResidualViT, which takes the previous I-feature along with the tokens of the current P-frame. A residual tokenizer—a simple linear layer—projects the I-feature into the ViT’s input space, while a token reduction strategy removes certain tokens from the P-frame. The reduced tokens and projected I-feature are then fed into the ViT, cutting down computational costs without compromising essential information.

The ResidualViT is trained with the feature distillation strategy, where the original ViT is a teacher network and the ResidualViT is a student network.
ResidualViT is trained using a feature distillation approach, where the original ViT acts as the teacher network and ResidualViT serves as the student network. Training is guided by cross-entropy loss based on the softmax similarity between frame and language features from a pretrained text encoder.
Extensive experiments demonstrate the efficiency of the proposed ResidualViT, showing substantial reductions in computational cost and significant improvements in inference speed.

**Strengths:**

**[S1]** The proposed method is a novel attempt to efficiently transfer a pretrained image model (vision-language model in practice) to video data.

**[S2]** This paper presents extensive experiments, including token reduction, input configuration, and distillation.

**Weaknesses:**

**[W1] Distillation through the soft target.**
In training, the proposed method leverages a language description corresponding to the input video to obtain soft targets by computing the softmax inner product between the visual and language features.
However, directly distilling the visual feature from the teacher network to the student network's feature can also be considered, as the pretrained vision and language encoder are kept frozen.
As shown in Section E, the ablation for the distillation strategy shows that there is no significant performance gap between models trained with each strategy.
Rather, the use of language descriptions appears to reduce the training efficiency due to the requirement of the CLIP text encoder.
An explanation and discussion of the benefits of distillation via soft target is required.

**[W2] Learning objective.**
The proposed learning objective makes the visual features from the teacher and student networks contain the same information. Meanwhile, correspondence between visual and language features seems to entirely rely on the pretrained model's capacity.
As the proposed ResidualViT focuses on the video-language task, considering frame-language interaction or video-language interaction in the learning objective would be good. (Note that this is simply a suggestion and the authors do not need to respond to this review.)

**[W3] Missing references.**
There are several previous works for efficient image-to-video transfer learning (e.g. [1-6]). While they have not focused on zero-shot natural language temporal video grounding, they could be included in Section 2.

```
[1] Lin et al., "Frozen CLIP models are efficient video learners," ECCV 2022
[2] Pan et al., "St-adapter: Parameter-efficient image-to-video transfer learning for action recognition," NeurIPS 2022
[3] Ni et al., "Expanding language-image pretrained models for general video recognition," ECCV 2022
[4] Ju et al., "Prompting visual-language models for efficient video understanding," ECCV 2022
[5] Yang et al., "AIM: Adapting image models for efficient video action recognition," ICLR 2023
[6] Park et al., "Dual-path adaptation from image to video transformers," CVPR 2023
```

**[W4] Additional issues.**
- Missing training details: In Table 2, the authors highlight the proposed method is not trained on downstream task datasets. However, the training data configuration has not appeared in the paper.
- Missing inference details: There is no explanation for inference, which should be included.
- Formatting: The reference formatting makes it hard for the reviewer to read and understand the paper. You may use \citep instead of \cite.

**Questions:**

**[Q1] Language description.**
- The proposed method requires language descriptions corresponding to the given videos. In that, the form of language data could affect the capacity of the learned representations and the final zero-shot natural language temporal video grounding performance. The quality of language description seems to be different for each training data. Is there an analysis according to the language descriptions' quality?
- Long-form videos are typically explained by multiple language descriptions as they contain multiple events. The proposed method takes a single text feature to compute the soft target, and I wonder whether this is sufficient when training a model using long-form videos.

**[Q2] Visual features.**
Are the visual features (i.e., I-/P-features) [CLS] token of the ViT's output?

**[Q3] Application.**
The proposed framework can be applied to any video or video-language tasks, such as action recognition and video-text retrieval.
Is there a reason to focus only on zero-shot natural language temporal video grounding?

---

> ### Author Response · Authors · 2024-11-21
>
> **[W1] Distillation through the soft target.** Thank you for pointing this out. We would like to emphasize that our work primarily addresses the practical challenge of efficiently deploying a trained model (forward inference), with training efficiency being outside the primary scope of this study. The result in Section E is encouraging, as it demonstrates that training can be conducted without the need for language annotations (reducing the training cost by approximately 10% for ViT-B/32, 3% for ViT-B/16 and 1.5% for ViT-L/14). Importantly, this observation does not diminish our main contribution: the effectiveness of learnable residual connections combined with token dropping during inference. If requested by the reviewer, we would be happy to revise the description of our secondary contribution (training via distillation) to clarify that the training process can be carried out with or without the use of language annotations.
>
> **[W2] Training Objective.** We thank the reviewer for this insightful suggestion. Incorporating frame-language or video-language interactions into the learning objective presents an exciting avenue for future research. We have included this suggestion in the manuscript to encourage the community to build upon our approach and explore further enhancements.
>
> In detail, we have added the following sentence to our conclusions: “We believe that our work opens up the possibility of extending the distillation objective to incorporate richer interactions between visual and language representations, as well as exploring additional large-scale pre-trained models that natively model temporal relationships.”
>
> **[W3] Missing references.** We thank the reviewer for highlighting these related works; each reference makes a great addition to our list of citations. In response, we have updated section 2 of the manuscript accordingly. We invite the reviewer to read the updated version of the manuscript. All additions have been highlighted in blue.
>
> **[W4] Additional issues.** We thank the reviewer for providing constructive feedback for improving the clarity of our work.
>
> 1. **Training details.** We include in “Appendix H Implementation details” the specifications of the training setting used in our manuscript. In particular, our method is trained on video-text pairs from the WebVid-2.5M dataset for five epochs. Our batch size is 2048 for ViT-B/32 and ViT-B/16 models and 1536 for ViT-L/14. At training time, we set the interleaving factor N_{train}=3. We use a constant learning rate of 0.0005 while weight decay and warmup are disabled. All model training is performed on 4 V100 GPUs, while inference only requires 1 V100 GPU. To improve the clarity of our manuscript, we modified the introductory paragraph of Section 4 to explicitly mention that the training details are provided in Appendix H. All changes to the manuscript are highlighted in blue.
> 2. **Inference details.** The inference process is detailed in Appendix C as part of the zero-shot grounding algorithm description. This section explains how video frames and textual queries are encoded into feature representations and then processed for the grounding task. Additionally, we provide the implementation details, such as FPS settings, the smoothing window, and the watershed threshold used, in Appendix H. We modified the introductory paragraph of Section 4 with a pointer to the inference details.
> 3. We appreciate the suggestion regarding citation formatting. We have updated the manuscript to use \citep{} for improved readability and consistency.
>
> **[Q1] Language description.**  We thank the reviewer for this question. We use the WebVid-2.5M dataset for distillation, which contains language-video pairs. As far as we are aware, there are no annotations for the quality of the language descriptions for this dataset, making an analysis of our approach according to the descriptions’ quality difficult. If the reviewer has suggestions for how to conduct such an analysis, we are happy to attempt it.
>
> Regarding long-form videos, we would like to clarify that we are not training using long-form videos. The dataset used for distillation is WebVid-2.5M whose average video duration is approximately 17.6 seconds. Additionally, in this work, we focus on frame-level feature embedding cost reduction. In this context, we apply the token reduction module for cost reduction and exploit the residual tokenizer for providing information from temporally neighbouring frames. Therefore, the approach is agnostic to the video duration, as our architecture remains a frame-based encoding backbone. Additionally, our zero-shot grounding baseline only exploits frame-text similarities for predicting relevant moments and is again agnostic to the video duration (the reason why we can adopt the same algorithm for short videos (Charades-STA / ActivityNe-Captions datasets) and long videos (MAD dataset)).

---

> > ### Author Response · Authors · 2024-11-21
> >
> > **[Q2] Visual features.** Yes, the visual features described in our work refer to the output [CLS] token. We updated Section 3 of the manuscript to clarify this aspect.
> >
> > **[Q3] Application.** We clarify that our approach is currently evaluated on two tasks: (i) Natural Language Temporal Video Grounding (our main focus) and (ii) Automatic Audio Descriptions.
> >
> > As requested, we conducted a new evaluation for the task of Action Recognition on the Kinetics-400 dataset, with detailed findings provided in Appendix L (pages 30-31 of the updated manuscript). **This experiment showcases that our ResidualViT features are transferable to tasks beyond those initially investigated.**
> >
> > Additionally, in response to the first question (Q1) raised by reviewer jGpQ,  we utilize our features in a fully supervised setting, selecting CG-DETR [A] as a representative of recent fully supervised grounding methods. We report a detailed analysis of our findings in Appendix K (page 30 of the updated manuscript) of the updated manuscript. **Once again, we show that ResidualViT features provide comparable performance to CLIP features at a lower average encoding cost, even in fully supervised settings.**
> >
> > Moreover, following on weakness 1 of reviewer cDMc we also evaluate our features in the task of Temporal Activity Localization (TAL), which does not make use of the language features. For this experiment, we select ActionFormer [B] as a representative of current state-of-the-art localization models, and we perform a comparative study on the performance obtained using CLIP features or ResidualViT features. Results are reported in Appendix M (page 31 of the updated manuscript). These results clearly **showcase the competitive performance of our features compared to the CLIP features in a visual-stream-only task that does not utilize the visual-language alignment.**
> >
> > Lastly, we would like to reiterate that our ResidualViT architecture is specifically designed to efficiently compute temporally dense sets of video features, as described in Lines 191-193 of the main paper. This approach is particularly suited for tasks requiring the computation of temporally dense visual representations, which often involve significant computational challenges due to the large volume of data being processed.
> >
> > In contrast, tasks such as text-to-video retrieval typically rely on sparse frame sampling (e.g., sampling 4 or 6 frames) and tend to reach performance saturation quickly (see Table 3d in [C]). The sparse sampling inherent to these tasks poses lower computational demands and makes them less suitable for our ResidualViT architecture, as the temporal gap between frames limits the utility of residual connections. Specifically, the residual connections are most effective when consecutive frames exhibit a degree of redundancy, as they provide valuable local information during the computation of P-features. Conversely, when frames are temporally distant, the residual connections tend not to capture meaningful local details, as explained in Lines 221-224 of the manuscript, which motivated our focus on dense tasks.
> >
> > Therefore, tasks that require (i) temporally dense features and (ii) multi-modal understanding, such as Natural Language Temporal Video Grounding (the primary focus of our work) and Automatic Audio Descriptions (discussed in Section 4.2), provide the most suitable testbed for our approach. However, as shown above, our approach is beneficial in other settings, including Action Recognition and Temporal Activity Localization.
> >
> > [A] Moon, WonJun, et al. "Correlation-guided query-dependency calibration in video representation learning for temporal grounding." arXiv preprint arXiv:2311.08835 (2023).
> >
> > [B] Zhang, Chen-Lin, Jianxin Wu, and Yin Li. "Actionformer: Localizing moments of actions with transformers." European Conference on Computer Vision. Cham: Springer Nature Switzerland, 2022.
> >
> > [C] Luo, Huaishao, et al. "Clip4clip: An empirical study of clip for end to end video clip retrieval." arXiv preprint arXiv:2104.08860 (2021).

---

> > ### Comment · Reviewer_y3Uz · 2024-11-25
> >
> > Thank you for your detailed responses and efforts. I have some additional questions
> > - How many frames were used during training? It must be written somewhere, but I can't find it. The use of a batch size of 2048 with 4 V100 GPUs seems somewhat unconvincing, particularly if the videos contain a large number of frames.
> > - Concerns regarding the title remain. The proposed method does not appear to be exclusively specialized for zero-shot natural language temporal video grounding, as evidenced by several presented results (Appendices F, K, and M). Additionally, the rationale for using the language description in [W1] has not yet been fully addressed, raising questions about the appropriateness of the title or its alignment with the key application focus.

---

> > > ### Author Response · Authors · 2024-11-25
> > >
> > > We thank the reviewer for the timely response. Here are our clarifications regarding the remaining concerns.
> > >
> > > **[Q4] Additional training information.** During training, our approach reasons over a small number of nearby frames (I-feature location and adjacent P-feature locations). In particular, the number of frames sampled for each video corresponds to the interleaving factor during training N_train=3 plus the corresponding I-features. This means that for each video we are sampling four frames. Note that the training WebVid-2.5M videos are stock videos and relatively short (average of 17.8 seconds), and we find that this sampling strategy suffices.
> > >
> > > The reported batch size (2048 samples = 2048 videos = 8192 frames) refers to the total batch size, each GPU accommodates 512 samples/videos for a total of 2048 frames. The GPU maximum memory allocation, as measured by the wandb log, is 13.5GB for the ViT-B/32 backbone. This low memory consumption is, of course, granted by the utilization of our ResidualViT module, which operates over fewer tokens and therefore is much more memory efficient as memory requirements scale quadratically with the number of tokens.
> > >
> > > We will include these additional details in “Appendix H Implementation details”.
> > >
> > >
> > > **[Q5] Title.** We appreciate the reviewer’s thoughtful feedback regarding the title. We recognize that the presented results, including those in Appendices F, K, and M, demonstrate the broader applicability of our proposed method beyond zero-shot natural language temporal video grounding (NLVTG). While this task remains the primary motivation and focus of our work, we understand the reviewer’s concern regarding the alignment of the title with the broader scope of our contributions.
> > >
> > > To better reflect the versatility of ResidualViT, we are open to suggestions for a revised version of the title. One option would be “**ResidualViT for efficient video encoding.**” However, we would greatly value the reviewer’s insights or specific recommendations for this adjustment. Additionally, if there are particular aspects that remain unclear, we would be happy to provide further clarification.

---

### Meta-Review · Area_Chair_7fRK · 2024-12-23

**Metareview:**

This paper presents a novel vision transformer architecture designed to efficiently compute frame-level features from videos by leveraging temporal redundancies. Its key technical contributions include the architectural design of ResidualViT and the introduction of a lightweight distillation strategy. The provided results well demonstrate the effectiveness and efficiency of the proposed method.

Overall, the reviewers appreciated the novelty, simplicity, and effectiveness of the proposed approach. However, several major concerns were raised. The most significant issue is that reviewers felt the paper’s presentation was somewhat disconnected --- after reading this manuscript, they tend to perceive this work more as a general approach utilizing video data and found little "truly effective" specialization is made for tailoring this method for zero-shot NLTVG tasks. The other major concerns include 1) more datasets need to be considered for validating the generalization of this method; 2) the comparisons could be unfair, given the use of the WebVid-2.5M dataset for distillation; 3) some related works need to be discussed; 4) more ablations are needed; and 5) some training and inference details need to be clarified.

The rebuttal is considered, which addresses some of these concerns. However, the biggest concernregarding the paper’s overall presentation remained unresolved for both Reviewer y3Uz and Reviewer jGpQ, who still viewed this method as a general approach utilizing video data rather than the one with specifically designed strategies for enhancing NLTVG task. The AC found this concern legitimate and agreed with reviewers that substantial revisions are required to address the structural and directional issues of the paper. Therefore, the final decision is rejection. The AC strongly encourages the authors to carefully reshape the whole presentation of this paper to prepare a more clear, compelling and targeted submission for future venues.

**Additional Comments On Reviewer Discussion:**

The major concerns are summarized in the meta-review above. While the rebuttal addresses some of these concerns, the biggest concern about the overall presentation of the paper remains. Both Reviewer y3Uz and Reviewer jGpQ continue to express significant concerns in this regard. As a result, the final decision is rejection.

---

### Decision · Program_Chairs · 2025-01-22

Reject